# Differences in the QBO response to stratospheric aerosol modification depending on injection strategy and species

Henning Franke[1,2], Ulrike Niemeier[1], and Daniele Visioni[3]

[1]Max Planck Institute for Meteorology, Bundesstr. 53, 20146 Hamburg, Germany
[2]International Max Planck Research School on Earth System Modelling, Bundesstr. 53, 20146 Hamburg, Germany
[3]Sibley School for Mechanical and Aerospace Engineering, Cornell University, Ithaca, NY, USA

**Correspondence:** Henning Franke (henning.franke@mpimet.mpg.de)

**Abstract.** A known adverse side effect of stratospheric aerosol modification (SAM) is the alteration of the quasi-biennial oscillation (QBO), which is caused by the stratospheric heating associated with an artificial aerosol layer. Multiple studies found the QBO to slow down or even completely vanish for point-like injections of $SO_2$ at the equator. The cause for this was found to be a modification of the thermal wind balance and a stronger tropical upwelling. For other injection strategies, different responses of the QBO have been observed. It has not yet been presented a theory which is able to explain those differences in a comprehensive manner. This is further complicated by the fact that the simulated QBO response is highly sensitive to the used model even under identical boundary conditions. Therefore, within this study we investigate the response of the QBO to SAM for three different injection strategies (point-like injection at the equator, point-like injection at 30° N and 30° S simultaneously, and areal injection into a 60° wide belt along the equator). Our simulations confirm that the QBO response significantly depends on the injection location. Based on the thermal wind balance, we demonstrate that this dependency is explained by differences in the meridional structure of the aerosol-induced stratospheric warming, i.e. the location and meridional extension of the maximum warming. Additionally, we also tested two different injection species ($SO_2$ and $H_2SO_4$). The QBO response is qualitatively similar for both investigated injection species. Comparing the results to corresponding results of a second model, we further demonstrate the generality of our theory as well as the importance of an interactive treatment of stratospheric ozone for the simulated QBO response.

## 1   Introduction

The modification of the stratospheric aerosol layer (SAM) by the artificial injection of sulfur dioxide ($SO_2$) into the lower stratosphere is currently widely discussed as a potential measure against global warming for the case of unmitigated greenhouse gas (GHG) emissions. It would basically mimic the processes after a large stratospheric volcanic eruption, resulting in an enhancement of the natural stratospheric sulfate aerosol layer. Since sulfate aerosols backscatter incoming short wave radiation (ISR), this enhancement of the stratospheric sulfate aerosol layer causes a negative radiative forcing (RF) onto the Earth system, which would counteract the tropospheric warming caused by increasing atmospheric GHG concentrations.

Besides backscattering ISR, sulfate aerosols also absorb parts of the outgoing tropospheric longwave radiation (OTLR) and the incoming near-infrared radiation (NIRR). The absorption of OTLR and NIRR causes a significant warming of the

lower tropical stratosphere (e.g., Heckendorn et al., 2009; Ferraro et al., 2011). This warming has important consequences for stratospheric dynamics, including the quasi-biennial oscillation (QBO). The QBO is a zonally symmetric oscillation of the zonal wind in the tropical stratosphere with an average period of approximately 28 months (Baldwin et al., 2001; Naujokat, 1986). It is characterized by an alternating downwelling of westerly and easterly winds from the upper stratosphere, above 5 hPa, into the tropopause region, where these wind patterns are rapidly attenuated (Baldwin et al., 2001; Holton, 2004). The

QBO has an impact on tropospheric winds (Garfinkel and Hartmann, 2011) and precipitation (Seo et al., 2013), as well as on the stratospheric transport to the extratropics (Plumb and Bell, 1982; Punge et al., 2009) and the polar vortex (Holton and Tan, 1980). After the major eruption of Mt. Pinatubo in June 1991, the lower stratosphere warmed by about 3 K, which led to a prolonged QBO westerly phase in the lower stratosphere (Labitzke, 1994), very likely due to an increased tropical upwelling induced by the aerosol warming (Giorgetta et al. (2011), pers. communication).

Multiple studies revealed that the QBO could also be heavily perturbed during a potential deployment of SAM (e.g., Aquila et al., 2014; Richter et al., 2017; Tilmes et al., 2018; Niemeier et al., 2020). For equatorial point injections, Aquila et al. (2014) obtained a prolonged or even permanent QBO westerly phase, depending on the injection rate. They attributed these modifications of the QBO basically to two physical mechanisms: a modification of the thermal wind balance due to the aerosol-induced warming of the lower tropical stratosphere, and an acceleration of the tropical upwelling as a response to this warming,

which decelerates the downward propagation of the QBO. Niemeier and Schmidt (2017) and Richter et al. (2017) further confirmed these results with other models.

    Together with equatorial point injections, a modification of the QBO has been also noticed for other injection strategies. Niemeier and Schmidt (2017) obtained a significantly prolonged westerly phase of the QBO for an injection into a zonal belt along the equator ranging from 30° N to 30° S with an injection rate of $10\,\mathrm{Tg(S)\,yr^{-1}}$, but weaker than for an equatorial point

injection with the same injection rate. For point-like injections in the extratropics, the QBO response to SAM is also different. Richter et al. (2017) showed that the QBO speeds up instead of slowing down for point-like injections at 15° N, 15° S, 30° N, and 30° S, testing an injection rate of $6\,\mathrm{Tg(S)\,yr^{-1}}$. The root cause of this acceleration was not finally determined, despite a detailed analysis of the 2° N - 2° S zonal mean momentum budget. Tilmes et al. (2018) analysed a simultaneous injection at two points at 15° N and 15° S for two different injection heights with injection rates of $12\,\mathrm{Tg(S)\,yr^{-1}}$ and $16\,\mathrm{Tg(S)\,yr^{-1}}$.

Within their simulations, the QBO slightly slows down, however, with a prolonged easterly phase within the lower stratosphere instead of a prolonged westerly phase as for equatorial point injections. They argue that the short simulation period and the low vertical resolution of their model may be a reason for these contradictory results.

    Additionally, Niemeier et al. (2020) showed that the simulated QBO response to SAM may be very sensitive to the used model itself by comparing two models (MAECHAM5-HAM and WACCM-110L) using the same model setup and injection

protocol. Both models showed a qualitatively similar QBO response on SAM, but quantitatively much stronger in WACCM-110L. The authors assumed differences in the vertical residual velocities in the tropics, also in a simulation without SAM, as the main cause of differences. Since the models used in the aforementioned studies as well as their specific setup vary significantly, the comparability of their results is consequently reduced. This further complicates the search for a comprehensive explanation of the at least partly contradictory QBO response to different injection locations.

To overcome this limitation, in this study we investigate the QBO response to three different injection locations for the same models as used by Niemeier et al. (2020), but with a different model setup in one case (see model description in Section 2). Both models followed the experiment protocol of the GeoMIP6 testbed experiment *accumH2SO4* (Weisenstein and Keith, 2018) to compare the different efficiency of $SO_2$ and $H_2SO_4$. Since multiple studies found that the forcing efficiency decreases significantly with increasing injection rates of $SO_2$ (e.g., Heckendorn et al., 2009; English et al., 2012; Niemeier and Timmreck,

2015; Vattioni et al., 2019), the direct injection of gaseous $H_2SO_4$ instead of $SO_2$ has been suggested as a potential alternative (Pierce et al., 2010; Benduhn et al., 2016). For both models we tested an injection into a zonal belt along the equator ranging from $30°$ N to $30°$ S and a simultaneous point-like injection at $30°$ N and $30°$ S, while for one model we additionally tested an equatorial point injection. Differently from previous studies, we aim for an advanced understanding of the dynamical mechanisms which lead to the SAM-induced modification of the QBO for different injection locations. We will in particular

focus on the modification of thermal wind balance by explicitly studying the SAM-induced modification of the meridional temperature gradient within the stratosphere, which was not done so far.

In Section 2, the models used in this study as well as the performed simulations are described. The results are structured as follows: In Section 3, we investigate the dependency of the QBO response to the injection location, rate, and species in our first model, MAECHAM5-HAM. Thereby, we give the theoretical explanation of the different response on SAM – focusing

on the modification of thermal wind balance – in Section 3.1.3. In Section 4, we then compare the SAM-induced modification of the QBO observed in MAECHAM5-HAM to that observed in CESM2(WACCM). This study ends with a discussion and a conclusion of the main findings in Section 5.

## 2   Model and setup of the simulations

### 2.1   MAECHAM5-HAM

MAECHAM5 is the middle atmosphere version of the spectral GCM ECHAM5 (Roeckner et al., 2003; Giorgetta et al., 2006; Roeckner et al., 2006). It simulates the evolution of atmospheric dynamics by numerically solving prognostic equations for temperature, surface pressure, vorticity, and divergence in terms of spherical harmonics. The different phases of water as well as tracers are transported within the model using a flux form semi-Lagrangian transport scheme (Lin and Rood, 1996). Details on ECHAM5 can be found in Roeckner et al. (2003). MAECHAM5 has a vertical domain which extends from the surface up to

$0.01\,\mathrm{hPa}$ while being resolved by 90 sigma-$p$ levels. Additionally, it accounts for the momentum flux deposition of unresolved gravity waves (GW) originating from the troposphere via a parameterization following Hines (1997a, b); its implementation into MAECHAM5 is described by Manzini et al. (2006). Therefore, MAECHAM5 internally generates a QBO in the tropical stratosphere (Giorgetta et al., 2006). For this study, MAECHAM5 was used with a spectral truncation at wave number 42 (T42) resulting in a horizontal Gaussian grid with $64 \times 128$ grid boxes with a size of $2.8° \times 2.8°$ per grid box.

MAECHAM5 was interactively coupled to the prognostic modal aerosol microphysical model HAM (Stier et al., 2005), which is based on the microphysical core M7 developed by Vignati et al. (2004). HAM calculates aerosol microphysical processes like nucleation, accumulation, condensation, and coagulation as well as the sulfate aerosol depletion via sedimentation

and deposition (Stier et al., 2005). In this setup of HAM, a simple stratospheric sulfur chemistry is applied in the stratosphere, which uses prescribed monthly oxidant fields and photolysis rates of, inter alia, ozone, OH, and $NO_x$ (Timmreck, 2001; Hommel and Graf, 2011). Therefore, the impact of SAM onto stratospheric ozone is not simulated within MAECHAM5-HAM. Within this stratospheric HAM version apart from the injected $SO_2$ or $H_2SO_4$, only natural sulfur emissions are taken into account. These simulations use the model setup described in Niemeier et al. (2009) and Niemeier and Timmreck (2015), where more details can be found. The HAM aerosol model couples back to the dynamics by the aerosol optical properties in the shortwave, longwave and near infrared range, which enter the radiative transfer scheme in MAECHAM5 and thus influence the temperature. Consequently, the interactive modification of the QBO is simulated within MAECHAM5-HAM, which will be hereafter referred to as ECHAM.

## 2.2 CESM2(WACCM)

The Community Earth System Model version 2 (release 2.1) in the Whole Atmosphere Community Climate model version CESM2(WACCM6) is a state of the art fully coupled climate model, used also in the new CMIP6 simulations (Gettelman et al., 2019). It uses 72 vertical layers up to about $150\,km$ and a $0.9°$ in latitude by $1.25°$ in longitude horizontal grid. WACCM6 includes convective, frontal, and orographic sources of GW, which propagate to drive the circulation of the middle atmosphere, including the QBO.

Whereas the standard version of WACCM6 uses comprehensive chemistry from the troposphere to the lower thermosphere, the version used here only simulates middle atmospheric (stratosphere, mesosphere and lower thermosphere) chemistry, with 98 simulated chemical species. Sulfate aerosols are treated using the Modal Aerosol Model (MAM4) as described in Liu et al. (2012, 2016), but with some modifications to change the mode widths and to the capabilities of sulfate aerosol to grow into the larger mode; an explanation of this and an evaluation of its capabilities in simulating volcanic aerosols after Pinatubo is given in Mills et al. (2016, 2017). CESM2(WACCM) will be hereafter referred to as CESM.

## 2.3 Simulations

The experimental setup of the simulations performed in this study is in accordance with the proposal of the GeoMIP6 testbed experiment *accumH2SO4* (Weisenstein and Keith, 2018) for both models. In all simulations, the sea surface temperature (SST) and the sea ice concentration (SIC) were set to monthly climatological values of the period 1988 to 2007 out of the AMIP SST data set following the experiment setup in Butchart et al. (2018). The GHG concentrations and the concentrations of ozone depleting substances (ODS) are taken from the SSP5-8.5 scenario of ScenarioMIP (O'Neill et al., 2016) for the year 2040. This combination of GHG and SST forcing allows to approximately mimic the surface cooling that would be produced by the sulfate layer, while having a consistent surface field for all models and thus removing the source of uncertainty derived from differences in the simulated cooling amongst models. Due to their coarse horizontal resolution, the used models are not able to simulate the rapid initial formation of accumulation mode sulfate particles $(AM-SO_4)$ after the injection of $H_2SO_4$. Therefore, the injection of $H_2SO_4$ is modeled as a direct injection of an $AM-SO_4$ population with a mode radius of $0.075\,\mu m$

and a standard deviation of 1.59 in ECHAM and a mode radius of 0.1 μm and a standard deviation of 1.5 in CESM, both following the proposal of *accumH2SO4* (Weisenstein and Keith, 2018).

With ECHAM, three different injection strategies have been simulated for both injection species, $SO_2$ and $AM-SO_4$: An injection into one single grid box centered at 1.4° N, 180° E (termed *point*), a simultaneous injection into two grid boxes centered at 29.3° N, 180° E and 29.3° S, 180° E (termed *2point*), and an injection into a zonally symmetric belt from 30° N to

30° S along the equator (termed *region*). The injections took place into three adjacent model layers in a height between 18 km and 20 km. With CESM, only the 2point and region injections have been simulated. The point injection strategy is not part of the *accumH2SO4* experiment protocol and was, therefore, not performed by CESM. For the 2point injections, the injections took place into a single model layer at 20 km, while for the region injections, the injections took place between 19 km and 21 km. All injection scenarios have been simulated with two different injection rates for both models: 5 and 25 $Tg(S) yr^{-1}$,

as given by the *accumH2SO4* protocol. The injection rate is the total amount of sulfur that is injected globally per year; for example, in the 2point injections with an injection rate of 25 $Tg(S) yr^{-1}$, each of both injection points has an injection rate of 12.5 $Tg(S) yr^{-1}$. For the 2point injection of $AM-SO_4$ with ECHAM, an additional simulation with an injection rate of 50 $Tg(S) yr^{-1}$ has been performed. An overview of all performed simulations and their setup can be found in Table 1.

All simulations were performed for a period of ten years. If not otherwise stated, the results presented in this study are

averaged over the last eight years of the respective simulation, since Visioni et al. (2019) showed that the artificial stratospheric sulfate layer has reached equilibrium already by the third year of deployment. All anomalies presented in this study have been calculated with respect to the control simulation (termed *contr-000*) of the corresponding model. The control simulations were performed with the same SST, SIC, GHG, and ODS specifications like the SAM simulations, but without any artificial injection of some sulfur species.

## 145 3 QBO response to SAM in ECHAM

ECHAM simulates the QBO well in the control simulation (Fig. 1 a-c), where it has a period of roughly 32 months, which is slightly longer than the observed period of approximately 28 months (Naujokat, 1986). Artificial sulfur injections may lead to a substantial modification of the QBO compared to the control simulation in ECHAM, depending on the injection strategy, injection species, and injection rate (Fig. 1 d-i). The equatorial point injections lead to the most significant modification of the

QBO compared to the other injection strategies: While an injection of $SO_2$ with an injection rate of 5 $Tg(S) yr^{-1}$ (Fig. 1 d) already leads to a drastic slowdown of the QBO with a prolonged lower stratospheric westerly phase, the QBO is locked in a constant lower stratospheric westerly phase for a $SO_2$ injection with an injection rate of 25 $Tg(S) yr^{-1}$ (Fig. 1 e). On top of the constant westerlies, constant easterlies are prevalent in the upper stratosphere. For the region injection of $SO_2$ with an injection rate of 5 $Tg(S) yr^{-1}$ (Fig. 1 g), the period of the QBO is clearly prolonged and westerlies dominate in the lower

stratosphere. For the region injection of $SO_2$ with an injection rate of 25 $Tg(S) yr^{-1}$ (Fig. 1 h), the QBO is also locked down in a permanent westerly phase, but with weaker westerlies than for the corresponding point injection. In contrast to the point

**Table 1.** Setup of all performed simulations. The point injections have been performed into a single equatorial grid box centered at $1.4^\circ$ N, $180^\circ$ E, the 2point injections have been performed into two boxes centered at $29.3^\circ$ N, $180^\circ$ E and $29.3^\circ$ S, $180^\circ$ E, and the region injections have been performed into a belt along the whole equator, ranging from $30^\circ$ N to $30^\circ$ S. The injection rate is the total amount of sulfur that is injected globally per year. Checkmarks indicate whether the experiment was performed for the according model, while values in brackets behind the checkmarks indicate the injection altitude.

| Experiment | Injection species | Injection rate | Injection location | ECHAM | CESM |
|---|---|---|---|---|---|
| contr-000 | - | - | - | ✓ | ✓ |
| point-so2-5 | $SO_2$ | $5\,\mathrm{Tg(S)\,yr^{-1}}$ | equatorial box | ✓ (18 - 20 km) | - |
| point-so2-25 | $SO_2$ | $25\,\mathrm{Tg(S)\,yr^{-1}}$ | equatorial box | ✓ (18 - 20 km) | - |
| point-so4-5 | $AM-SO_4$ | $5\,\mathrm{Tg(S)\,yr^{-1}}$ | equatorial box | ✓ (18 - 20 km) | - |
| point-so4-25 | $AM-SO_4$ | $25\,\mathrm{Tg(S)\,yr^{-1}}$ | equatorial box | ✓ (18 - 20 km) | - |
| 2point-so2-5 | $SO_2$ | $5\,\mathrm{Tg(S)\,yr^{-1}}$ | 2 boxes | ✓ (18 - 20 km) | ✓ (20 km) |
| 2point-so2-25 | $SO_2$ | $25\,\mathrm{Tg(S)\,yr^{-1}}$ | 2 boxes | ✓ (18 - 20 km) | ✓ (20 km) |
| 2point-so4-5 | $AM-SO_4$ | $5\,\mathrm{Tg(S)\,yr^{-1}}$ | 2 boxes | ✓ (18 - 20 km) | ✓ (20 km) |
| 2point-so4-25 | $AM-SO_4$ | $25\,\mathrm{Tg(S)\,yr^{-1}}$ | 2 boxes | ✓ (18 - 20 km) | ✓ (20 km) |
| 2point-so4-50 | $AM-SO_4$ | $50\,\mathrm{Tg(S)\,yr^{-1}}$ | 2 boxes | ✓ (18 - 20 km) | - |
| region-so2-5 | $SO_2$ | $5\,\mathrm{Tg(S)\,yr^{-1}}$ | $30^\circ$ N to $30^\circ$ S | ✓ (18 - 20 km) | ✓ (19 - 21 km) |
| region-so2-25 | $SO_2$ | $25\,\mathrm{Tg(S)\,yr^{-1}}$ | $30^\circ$ N to $30^\circ$ S | ✓ (18 - 20 km) | ✓ (19 - 21 km) |
| region-so4-5 | $AM-SO_4$ | $5\,\mathrm{Tg(S)\,yr^{-1}}$ | $30^\circ$ N to $30^\circ$ S | ✓ (18 - 20 km) | ✓ (19 - 21 km) |
| region-so4-25 | $AM-SO_4$ | $25\,\mathrm{Tg(S)\,yr^{-1}}$ | $30^\circ$ N to $30^\circ$ S | ✓ (18 - 20 km) | ✓ (19 - 21 km) |

and region injections, the QBO is basically not modified for the 2point injections of both tested injection rates in terms of periodicity and strength with respect to the control simulation (Fig. 1 j,k).

For an injection of $AM-SO_4$ (Fig. 1 right), the modification of the QBO is slightly stronger than for the corresponding injection of $SO_2$ with the same injection strategy and rate (Fig. 1 middle) when using the point and region injection strategy. For the 2point injections, the strength of the QBO modification does not show a significant dependence on the injection species in our simulations.

### 3.1 Dynamic mechanisms of QBO modification

The dynamic mechanisms which cause the observed modification and breakdown of the QBO for an equatorial point injection of $SO_2$ have been investigated by Aquila et al. (2014). They argue that the absorption of radiation in the near IR and terrestrial wavelengths by the artificial sulfate aerosols and the associated lower stratospheric heating are the root cause of the observed changes in QBO dynamics. In more detail, they identified that this aerosol-induced warming modifies thermal wind balance in the lower tropical stratosphere and increases the residual tropical upwelling in the rising branch of the Brewer-Dobson circulation (BDC), both causing a modification of the QBO.

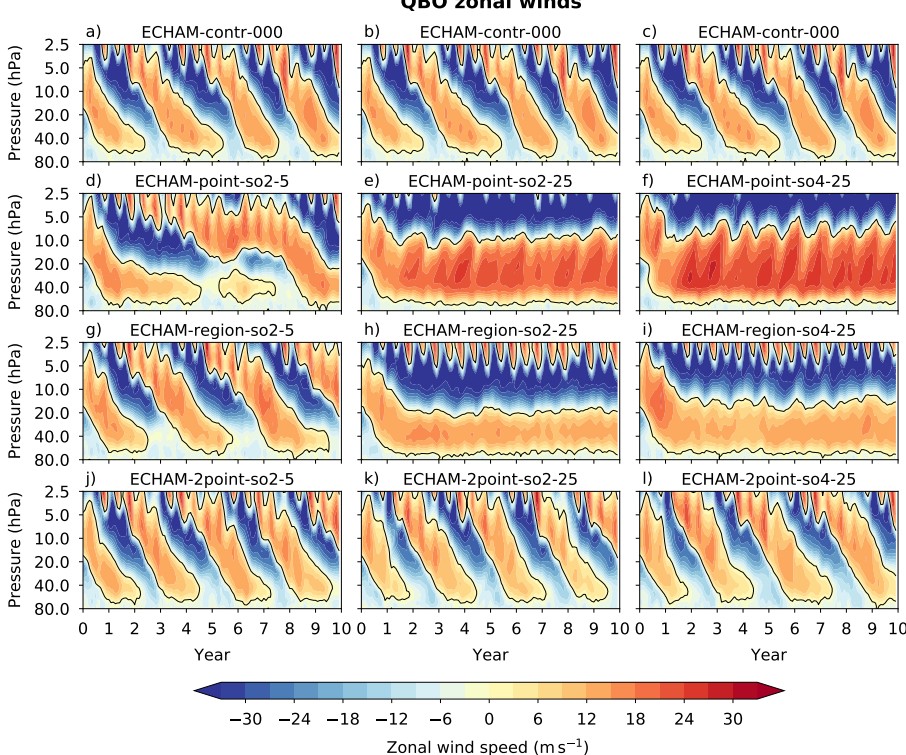

**Figure 1.** Time-height cross sections of the $5°$ N to $5°$ S mean zonal wind in the stratosphere over the simulation period of ten years for ECHAM for different injection scenarios. The columns indicate the injection species and rate: The left column shows $SO_2$ injections with an injection rate of $5\,Tg(S)\,yr^{-1}$, the middle column shows $SO_2$ injections with an injection rate of $25\,Tg(S)\,yr^{-1}$, and the right column shows $AM−SO_4$ injections with an injection rate of $25\,Tg(S)\,yr^{-1}$. The rows indicate the injection strategy: The 1[st] row shows the control simulation, the 2[nd] row shows the point injection, the 3[rd] row shows the region injections and the 4[th] row shows the 2-point injection. The solid black line marks a tropical mean zonal wind speed of $0\,m\,s^{-1}$.

In this Section, we will investigate the reasons for the different QBO response to the three tested injection strategies exemplarily based on an injection of $SO_2$ with an injection rate of $25\,Tg(S)\,yr^{-1}$ (experiments point-so2-25, region-so2-25, 2point-so2-25). This injection scenario follows the experimental setup of Aquila et al. (2014) with regard to the injection type and has a high signal-to-noise ratio due to the high injection rate. The impact of a lower injection rate and another injection species (i.e. $AM−SO_4$ instead of $SO_2$) will be discussed in Sections 3.2 and 3.3.

Additionally, we are aware of the fact that the QBO may also change due to a modified wave driving. However, we found no significant changes in QBO wave driving in our simulations (not shown), which is in agreement with earlier studies (e.g. Aquila et al., 2014; Richter et al., 2017; Tilmes et al., 2018). Therefore, within this Section we will only focus on the increase of the tropical upwelling and the modification of thermal wind balance.

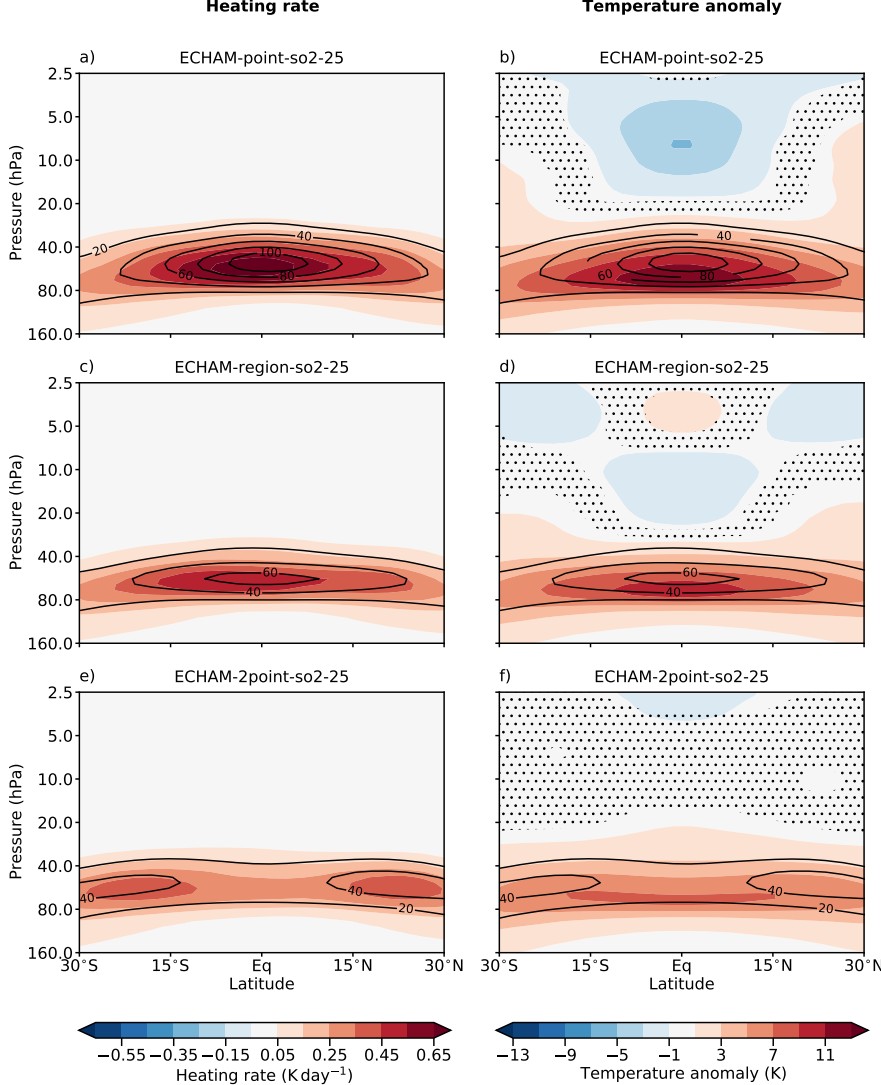

**Figure 2.** Latitude-height cross section of the zonal mean net aerosol heating rate (left column) and of the anomaly of the zonal mean temperature $\overline{T}$ (right column) for the ECHAM simulations of point-so2-25 (top row), region-so2-25 (middle row), and 2point-so2-25 (bottom row). Stippling indicates areas where the temperature anomalies are not significant at the 95 % level in a student's $t$-test. Black contour lines indicate the anomaly of the zonal mean sulfate mass mixing ratio $\overline{m}_{SO4}$ in intervals of $20\,\mu g(S)\,kg^{-1}$.

### 3.1.1 Aerosol-induced heating of the lower stratosphere

The artificial sulfate aerosols heat the lower stratosphere by the absorption of OTLR and NIRR, whereby the location and magnitude of this heating strongly correlate with those of the sulfate mass mixing ratio $\overline{m}_{SO4}$ (Fig. 2 a,c,e). For an equatorial

point injection (Fig. 2 a), the sulfate aerosols are strongly concentrated within the tropics, which leads to a strong heating of the lower tropical stratosphere peaking at the equator. In contrast, the sulfate aerosols are distributed meridionally more uniform for the region injection and even more so for the 2point injection (Fig. 2 c,e), which also results in a meridionally more uniform heating for the region and the 2point injection than for the point injection.

This aerosol-induced heating results in a significant positive temperature anomaly centered at the equator (Fig. 2 b,d,f). Following the meridional structure of the net aerosol heating rates, the lower stratospheric temperature anomaly has a clear equatorial peak for the point injection and its poleward gradients are sharp (Fig. 2 b). For the region injection, the lower stratospheric temperature anomaly still peaks at the equator, but with a smaller absolute magnitude; leading to a smaller poleward gradient (Fig. 2 d). For the 2point injection, the temperature anomaly is meridionally nearly uniform between $15^\circ$ N and $15^\circ$ S (Fig. 2 f).

The warming of the lower stratosphere is the primary perturbation induced by the sulfate aerosols, as indicated by the good agreement of the sulfate mass mixing ratio, the net aerosol heating rates, and the temperature anomalies. All changes in dynamics – including the QBO – are obviously induced by this initial warming in a second step.

Opposite to the lower stratospheric warming, statistically significant negative temperature anomalies are located in the middle and upper tropical stratosphere for all three injection strategies (Fig. 2 b,d,f). However, Figure 2 clearly shows that this cooling is not induced by the radiative effects of the aerosols as it is located well above the aerosol layer and does not match with the net aerosol heating rates. Consequently, these negative temperature anomalies have been induced dynamically due to an increased tropical upwelling (see Aquila et al., 2014). Therefore, they cannot be seen as a root cause of any change in the QBO.

### 3.1.2 Modification of the residual circulation

Following Aquila et al. (2014), an increase of the tropical upwelling in the rising branch of the Brewer-Dobson circulation (BDC) due to the aerosol-induced warming is the main reason for the modification of the QBO. Commonly, the BDC is treated in the so-called Transformed Eulerian Mean (TEM) framework as outlined by Andrews et al. (1987), in which it is represented by the residual mean circulation. The residual mean circulation itself may be described by the residual meridional and vertical velocity $\overline{v}^*$ and $\overline{w}^*$, respectively, or by its mass stream function $\chi$. For equatorial point injections of $SO_2$, Aquila et al. (2014) showed that an aerosol-induced increase of $\overline{w}^*$ is associated with a stronger residual vertical advection of zonal momentum $(-\overline{w}^*\overline{u}_z)$. A stronger $-\overline{w}^*\overline{u}_z$ in the tropical stratosphere weakens the downwelling of the QBO phases, which leads to a prolongation of the QBO period.

Our simulations confirm that $\overline{w}^*$ increases statistically significantly within the tropics for point-so2-25 and region-so2-25 and that this increase results in a stronger $-\overline{w}^*\overline{u}_z$ in the upper tropical stratosphere (Fig. 3 a,b). Thereby, the anomalies are slightly stronger for point-so2-25 than for region-so2-25 due to the stronger aerosol-induced stratospheric warming. The maximum anomalies of $\overline{w}^*\overline{u}_z$ are located at the altitudes of strongest easterly shear (see Fig. 1 e,h). This indicates that the increase of the tropical upwelling helps to maintain the permanent westerlies against the downwelling easterlies aloft. For 2point-so2-25, $\overline{w}^*$ as well as $-\overline{w}^*\overline{u}_z$ do not show a statistically significant increase throughout the whole tropical stratosphere (i.e. between $15^\circ$ N

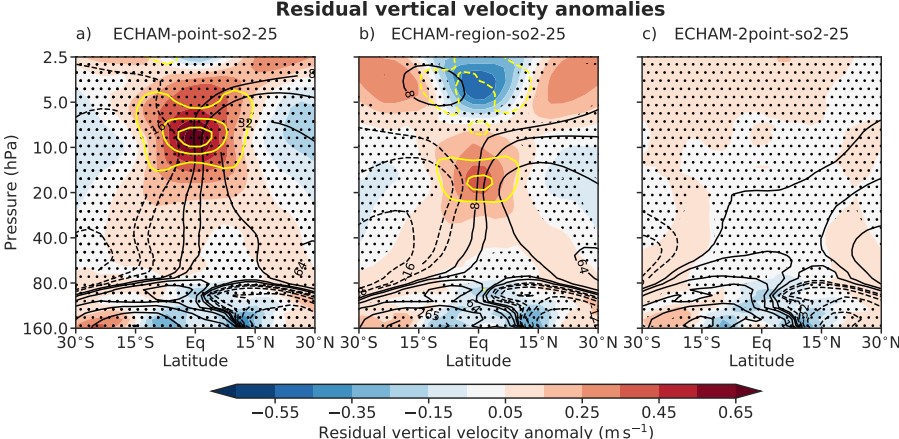

**Figure 3.** Latitude-height cross section of the anomaly of the zonal mean residual vertical velocity $\overline{w}^*$ for the ECHAM simulations of point-so2-25 (a), region-so2-25 (b), and 2point-so2-25 (c). Stippling indicates areas where anomalies are not significant at the 95 % level in a student's *t*-test. Black contour lines show the anomaly of the zonal mean residual mass stream function $\chi$ in $\mathrm{kg\,s^{-1}}$. The thin solid contours indicate a clockwise circulation anomaly and the dashed contours indicate an anti-clockwise circulation anomaly. The contour interval is logarithmic starting at $8\,\mathrm{kg\,s^{-1}}$ and $-8\,\mathrm{kg\,s^{-1}}$, respectively, while the $0\,\mathrm{kg\,s^{-1}}$ contour is omitted. Yellow contour lines denote the residual vertical advection of zonal momentum $-\overline{\omega}^*\overline{u}_z$ with a contour interval of $0.3\,\mathrm{m\,s^{-1}\,day^{-1}}$, starting at $0.15\,\mathrm{m\,s^{-1}\,day^{-1}}$ and $-0.15\,\mathrm{m\,s^{-1}\,day^{-1}}$. The solid lines indicate a positive anomaly, the dashed lines indicate a negative anomaly, and the $0\,\mathrm{m\,s^{-1}\,day^{-1}}$ contour is omitted.

and 15° S) (Fig. 3 c). The zonal mean residual circulation as a whole is also only weakly modified in the tropical stratosphere. This is in accordance with our observation that the amplitude as well as the periodicity of the QBO basically remain unchanged for 2point-so2-25 (Fig. 1 k).

The reason for the increase of $\overline{w}^*$ is the aerosol-induced stratospheric temperature anomaly, which alters the characteristics
of the zonal jets in the extratropical stratosphere. Thereby, the conditions for the vertical propagation of planetary waves in this region change. As a consequence, the extratropical wave-driving of the residual mean circulation increases, which ultimately speeds up the whole BDC. This mechanism has been investigated by Tilmes et al. (2018) for SAM simulations and was also recognized in simulations of a tropical volcanic eruption by Bittner et al. (2016).

In the upper stratosphere (i.e. between $20\,\mathrm{hPa}$ and $3\,\mathrm{hPa}$ in point-so2-25 and between $25\,\mathrm{hPa}$ and $8\,\mathrm{hPa}$ in region-so2-25),
this increase of $\overline{w}^*$ is superimposed by changes of the secondary meridional circulation (SMC) of the modified QBO itself. During a permanent QBO westerly phase, the SMC would also be permanently locked in its corresponding "westerly" phase, which acts to increase $\overline{w}^*$ within the tropical stratosphere (Plumb and Bell, 1982). Our experiments indicate that a large fraction of the increase of $\overline{w}^*$ in the upper tropical stratosphere (Fig. 3 a,b) can be attributed to this "indirect" acceleration via the SMC, especially in the upper stratosphere. For example, in the experiment point-so2-5, the tropical $\overline{w}^*$ increased by up to 100% in
the upper stratosphere (not shown). In contrast, in ECHAM simulations with permanent lower stratospheric easterlies instead

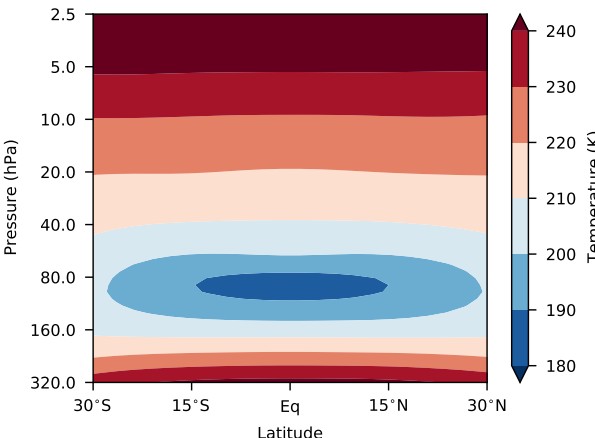

**Figure 4.** Latitude-height cross section of the zonal mean temperature $\overline{T}$ in contr-000

of a QBO, Niemeier et al. (2011) obtained an increase of the tropical $\overline{w}^*$ of only 5 to 10% for an equatorial point injection of $SO_2$ with an injection rate of $4\,\mathrm{Tg(S)\,yr^{-1}}$. Therefore, one has to be cautious when interpreting the strong positive anomalies of $-\overline{w}^*\overline{u}_z$ observed in the upper stratosphere in point-so2-25 and region-so2-25 as the primary cause for the disruption of the QBO since they are – at least partly – rather its consequence.

Within the TEM framework, the characteristics of the general acceleration of the BDC can be further directly linked to the tropical confinement of the aerosol-induced temperature anomaly, as shown by Dunkerton (1983) in a study on the effect of the 1963 eruption of Mt. Agung on the QBO. Following the equation of the residual mean meridional streamfunction (Plumb and Bell, 1982), only a tropically confined heating modifies the BDC, whereas a meridionally uniform heating has no effect. This explains why the residual tropical upwelling increases in point-so-25 and region-so2-25, but not in 2point-so2-25: While in point-so2-25 and region-so2-25 the aerosol heating has an equatorial peak and decreases rather sharply towards the extratropics, in 2point-so2-25 it is meridionally nearly uniform within the tropics (see Sec. 3.1.1).

Consequently, it is ultimately the meridional shape of the aerosol-induced lower stratospheric temperature anomaly, or – simply spoken – the degree of tropical confinement of the artificial sulfate aerosols, what determines the QBO response to artificial sulfate injections.

### 3.1.3 Modification of thermal wind balance

Besides via an increase of the tropical upwelling in the rising branch of BDC, Aquila et al. (2014) also attributed the observed changes in the QBO to a modification of thermal wind balance. Thermal wind balance is an atmospheric state equation, which links the zonal mean meridional temperature gradient $\overline{T}_y$ to the zonal mean vertical wind shear $\overline{u}_z$. It is defined as

$$\overline{u}_z = -R(H\beta y)^{-1}\overline{T}_y \tag{1}$$

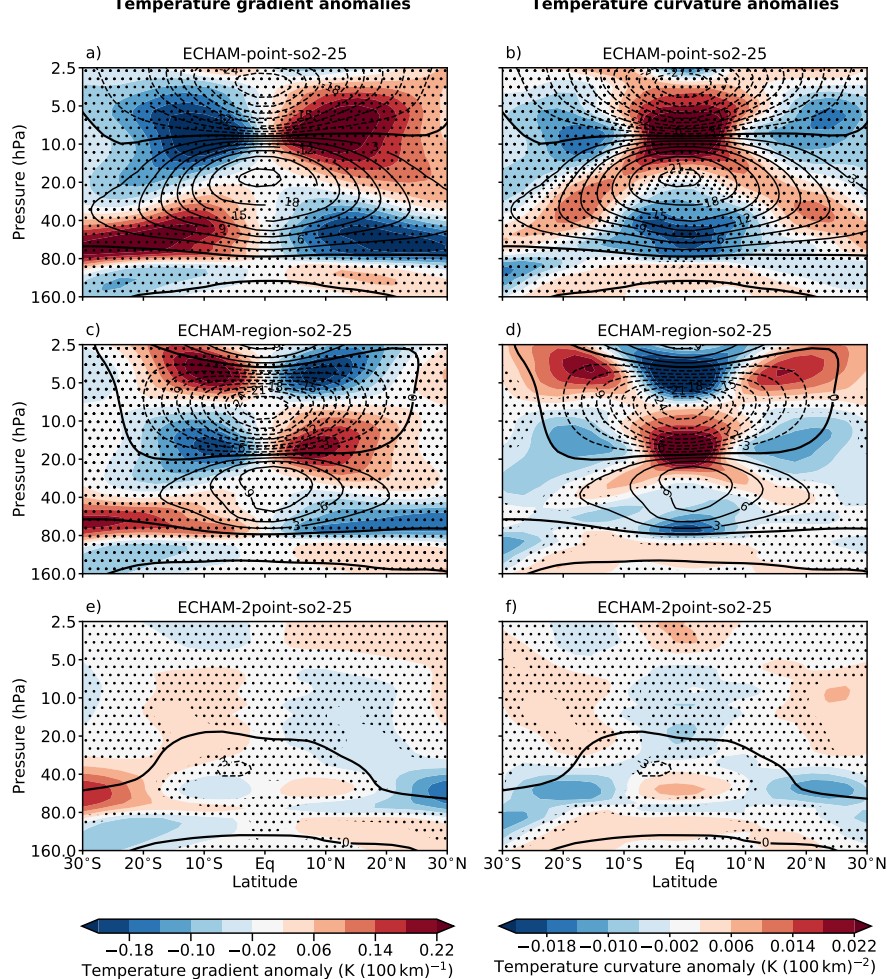

**Figure 5.** Latitude-height cross section of the anomaly of the meridional zonal mean temperature gradient $\overline{T}_y$ (left column) and of the anomaly of the meridional zonal mean temperature curvature $\overline{T}_{yy}$ (right column) for the ECHAM simulations of point-so2-25 (top row), region-so2-25 (middle row), and 2point-so2-25 (bottom row). Stippling indicates areas where anomalies are not significant at the 95 % level in a student's $t$-test. Black contour lines indicate the anomaly of the zonal mean zonal wind speed $\overline{u}$ in intervals of $3\,\mathrm{m\,s^{-1}}$, with the thick black line denoting $\overline{u} = 0\,\mathrm{m\,s^{-1}}$. Solid lines denote a westerly anomaly, dashed lines denote an easterly anomaly.

250 for an equatorial $\beta$-plane (Holton, 2004). Assuming equatorial symmetry of the zonal mean temperature field, one can set $\overline{T}_y = 0\,\mathrm{K\,km^{-1}}$ at the equator and apply the rule of L'Hospital (Holton, 2004):

$$\overline{u}_z = -R(H\beta)^{-1}\overline{T}_{yy}. \tag{2}$$

Within Equation 1 and 2, $R$ denotes the gas constant for dry air, $H$ the scale height, and $\beta$ the meridional gradient of the Coriolis parameter at the equator. $\overline{T}_{yy}$ denotes the meridional curvature of the zonal mean temperature.

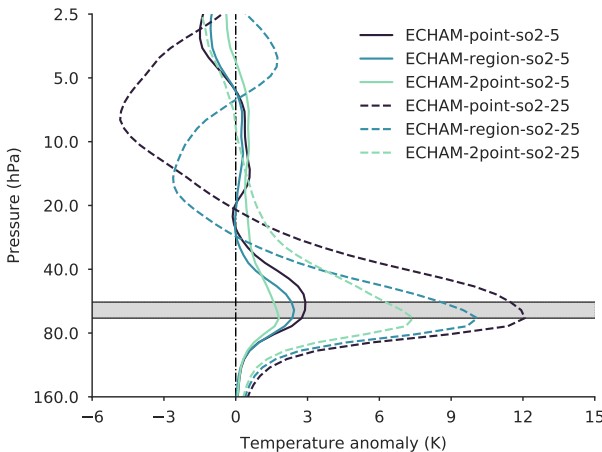

**Figure 6.** Vertical profile of the $5°$ N to $5°$ S mean temperature anomaly for the 5 and $25\,\mathrm{Tg(S)}\,\mathrm{yr}^{-1}$ injections simulated with ECHAM. The horizontal grey bar marks the injection layer. The vertical dashed-dotted line marks a temperature anomaly of $0\,\mathrm{K}$.

However, despite QBO changes due to artificial sulfur injections are frequently interpreted as the consequence of an increased residual tropical upwelling and a modification of thermal wind balance, one can't see both as two separate processes. In contrast, the acceleration of the BDC discussed in Section 3.1.2 is rather the mechanism by which thermal wind balance is reestablished for the aerosol-induced lower stratospheric temperature anomaly (Holton, 2004). While the differences in the increase of $\overline{w}^*$ directly explain the different QBO response to different injection strategies in first order (see Sec. 3.1.2), in this Section we show that the differences in the QBO response can be also linked directly to the meridional shape of the aerosol-induced lower stratospheric temperature anomaly via thermal wind balance.

As discussed in Section 3.1.1, an equatorial point injection results in a significant positive temperature anomaly centered at the equator (Fig. 2 b). In the climatological mean – without artificial sulfur injections and represented by contr-000 – the lower tropical stratosphere is much colder than the lower midlatitudinal stratosphere (Fig. 4), leading to a poleward $\overline{T}_\mathrm{y}$ that is positive. The aerosol-induced warming abates the poleward $\overline{T}_\mathrm{y}$ (Fig. 5 a) within the lower tropical stratosphere between $40\,\mathrm{hPa}$ and $80\,\mathrm{hPa}$, which is accompanied by a significant negative anomaly of $\overline{T}_\mathrm{yy}$ centered at the equator (Fig. 5 b). Following Equation 2, a negative anomaly of $\overline{T}_\mathrm{yy}$ results in stronger westerly shear. Consequently, a strong westerly anomaly of the zonal mean zonal wind $\overline{u}$ is located on top of the injection layer in order to maintain thermal wind balance (Fig. 5 b). This results in the observed constant westerly QBO phase (Fig. 1 b).

For region-so2-25, the QBO was also found to be locked in a permanent westerly phase, but the vertical extent as well as the strength of the westerlies is weaker than for point-so2-25, which is in agreement with the results of Niemeier and Schmidt (2017). The reason for the weaker westerlies is the meridionally more uniform temperature anomaly within the lower tropical stratosphere (Fig. 2 d). Therefore, the strongest modifications of $\overline{T}_\mathrm{y}$ in the lower stratosphere are located poleward of approximately $20°$ N and $20°$ S, while its modification close to the equator is relatively small (Fig. 5 c). Accordingly, also the

negative anomaly of $\overline{T}_{yy}$ and – following thermal wind balance (Eq. 2) – the induced anomaly of westerly shear is weaker near the equator compared to point-so2-25 (Fig. 5 d). Consequently, the lower stratospheric westerly anomaly of $\overline{u}$ is weaker for region-so2-25 than for point-so2-25.

For 2point-so2-25, the QBO was not found to be modified significantly and it basically preserved its natural periodicity (Fig. 1 k). Due to the extratropical injection locations, the highest sulfate concentrations are located at approximately $20°$ N and $20°$ S and so is the associated heating (Fig. 2 e). Therefore, the lower-stratospheric temperature anomaly is meridionally nearly uniform between approximately $15°$ N and $15°$ S (Fig. 2 f). Consequently, $\overline{T}_y$ as well as $\overline{T}_{yy}$ are only very weakly modified between $40\,\mathrm{hPa}$ and $80\,\mathrm{hPa}$ and between approximately $15°$ S and $15°$ N (Fig. 5 e,f) and these anomalies are hardly statistically significant. Following thermal wind balance, this implies that zonal wind anomalies in the lower stratosphere are small as well (Fig. 5 e,f), which is in accordance with the QBO in principle remaining in its natural shape.

As discussed in Section 3.1.1, the negative temperature anomalies above approximately $20\,\mathrm{hPa}$ have been induced dynamically due to an increased tropical upwelling and negative temperature advection (see Aquila et al., 2014). While the corresponding anomalies of $\overline{T}_y$ and $\overline{T}_{yy}$ above $20\,\mathrm{hPa}$ are of course still in thermal wind balance with the upper stratospheric wind field (Fig. 5), it is important to mention that this agreement does not imply causality in the way that these upper stratospheric temperature anomalies have caused the lower stratospheric westerly anomaly and QBO modification.

Our results clearly show that differences in the QBO response with respect to our three tested injection strategies are linked to differences in the meridional structure of the aerosol-induced temperature anomaly. Therefore, the absolute strength of the aerosol-induced lower stratospheric temperature anomaly does not permit a statement about the strength of the QBO modification when comparing different injection strategies. For instance, the tropical (i.e. $5°$ N to $5°$ S) mean temperature anomaly within the injection layer is more than twice as high in 2point-so2-25 than in point-so2-5 (Fig. 6). However, the QBO is heavily perturbed in point-so2-5, while for 2point-so2-25 it remains nearly unchanged (Fig. 1 d,k). This comparison shows that within

$$\overline{u}_z \sim R(H\beta)^{-1} L^{-2} \overline{T}, \tag{3}$$

which is often used as an approximation of thermal wind balance for QBO variations centered at the equator (Baldwin et al., 2001), the scaling factor L depends on the injection strategy. Therefore, Equation 3 cannot be used when comparing the QBO response to different injection strategies.

Since it is the degree of tropical confinement of the artificial sulfate aerosols what is ultimately decisive for the observed QBO response also when explaining the observed QBO changes solely as the consequence of an increased residual tropical upwelling, we will use thermal wind balance in our argumentation throughout this study as it directly links the observed QBO changes to the observed aerosol-induced temperature anomalies.

## 3.2 Impact of injection rate

For the point and the region injection strategy, the QBO was found to be impacted much less in our experiments with an injection rate of $5\,\mathrm{Tg(S)\,yr^{-1}}$ than in those with an injection rate of $25\,\mathrm{Tg(S)\,yr^{-1}}$ and it basically maintained its oscillating

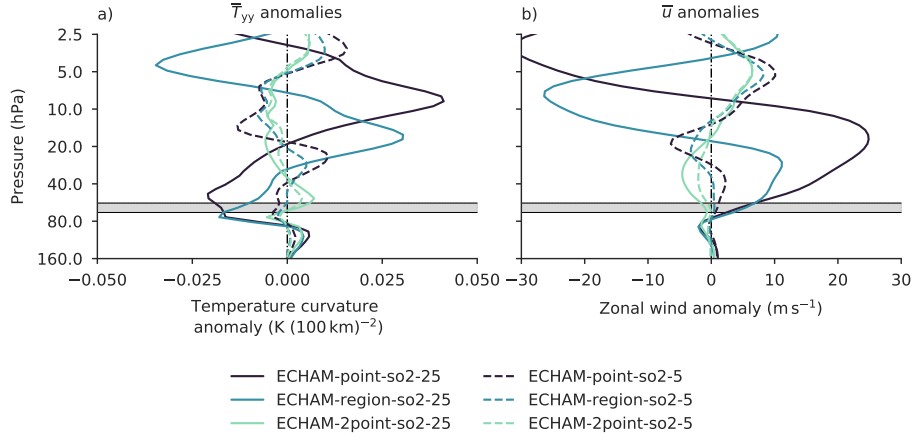

**Figure 7.** Vertical profile of the $5°$ N to $5°$ S mean anomaly of the temperature curvature $\overline{T}_{yy}$ (a) and the tropical mean anomaly of the zonal wind $\overline{u}$ (b) for the ECHAM simulations of a $SO_2$ injection with an injection rate of $5\,Tg(S)\,yr^{-1}$ (dashed) and $25\,Tg(S)\,yr^{-1}$ (solid). The horizontal grey bar marks the injection layer. The vertical dashed-dotted line marks an anomaly of $0\,K\,(100\,km)^{-2}$ (a) and $0\,m\,s^{-1}$ (b).

behaviour (Fig. 1 d,g). This is explained by the clearly lower tropical sulfate burden, which results out of the lower injection rate. The sulfate burden determines the strength of the lower stratospheric heating by absorption of OTLR and NIRR. Accord-

310 ingly, the tropical mean temperature anomalies are clearly weaker in our experiments with an injection rate of $5\,Tg(S)\,yr^{-1}$ than in those ones with an injection rate of $25\,Tg(S)\,yr^{-1}$ (Fig. 6). In contrast, the temperature anomaly in the extratropical stratosphere is rather independent of the injection rate for all injection strategies (not shown), since absorptive heating is generally weak in this region due to low values of OTLR and NIRR. Therefore, $\overline{T}_y$ changes much less for a lower injection rate. The tropical mean anomalies of $\overline{T}_{yy}$ in and slightly above the injection layer are clearly smaller and vertically less extended for an

315 injection rate of $5\,Tg(S)\,yr^{-1}$ compared to $25\,Tg(S)\,yr^{-1}$ (Fig. 7 a). Following Equation 2, this results in significantly smaller anomalies of the tropical mean zonal wind within the lower stratosphere (Fig. 7 b). For the point and the region injection strategy, the strength of the westerly anomaly is reduced by a factor of $\sim 10$ when reducing the injection rate from $25\,Tg(S)\,yr^{-1}$ to $5\,Tg(S)\,yr^{-1}$ and, consequently, the QBO is not locked in a permanent westerly phase (Fig. 1 d,g). For the 2point injection strategy, the tropical mean anomaly of $\overline{T}_{yy}$ is small for both tested injection rates (Fig. 7 a). Accordingly, the QBO was found

not to be modified significantly for either tested injection rates when applying the 2point injection strategy.

### 3.3 Impact of injection species

For all three tested injection strategies, the response of the QBO is in principle independent of the injection species – $SO_2$ or $AM-SO_4$ – in our experiments with an injection rate of $25\,Tg(S)\,yr^{-1}$ (Fig. 1). This is reasonable, since the meridional distribution of the artificial sulfate aerosols, which can be seen as a proxy for the strength of the lower stratospheric temperature

anomaly, does in principle exhibit the same shape for both tested injection species except for different absolute values (Fig. 8).

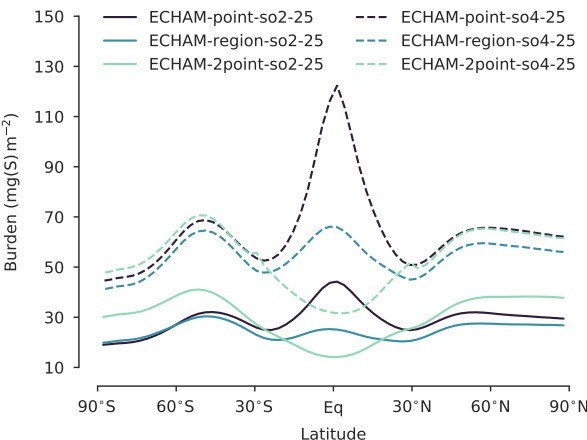

**Figure 8.** Zonal mean artificial sulfate burden for the ECHAM simulations of the $SO_2$ injections (dashed lines) and the $H_2SO_4$ injections (solid lines) with an injection rate of $25\,\mathrm{Tg(S)\,yr^{-1}}$ as a function of latitude.

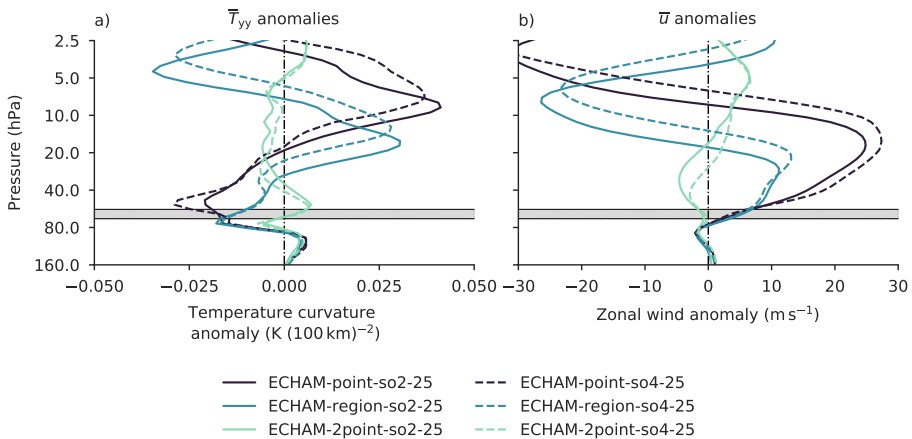

**Figure 9.** Vertical profile of the $5°$ N to $5°$ S mean anomaly of the temperature curvature $\overline{T}_{yy}$ (a) and the tropical mean anomaly of the zonal wind $\overline{u}$ (b) for the ECHAM simulations of injections with an injection rate of $25\,\mathrm{Tg(S)\,yr^{-1}}$. The horizontal grey bar marks the injection layer. The vertical dashed-dotted line marks an anomaly of $0\,\mathrm{K}\,(100\,\mathrm{km})^{-2}$ (a) and $0\,\mathrm{m\,s^{-1}}$ (b).

We showed that the modification of the QBO depends clearly on the meridional structure of the stratospheric temperature anomaly and is rather independent of its absolute value (Sec. 3.1.3).

However, for the point and region injection strategy, the modification of the QBO was found to be slightly stronger with respect to the strength and the vertical extent of the lower stratospheric westerlies when injecting $AM-SO_4$ instead of $SO_2$ 330 based on our experiments with an injection rate of $25\,\mathrm{Tg(S)\,yr^{-1}}$ (Fig. 1). This is a consequence of the in general higher sulfate burden, which results from an injection of $AM-SO_4$ compared to an injection of $SO_2$. As described in Section 3.2,

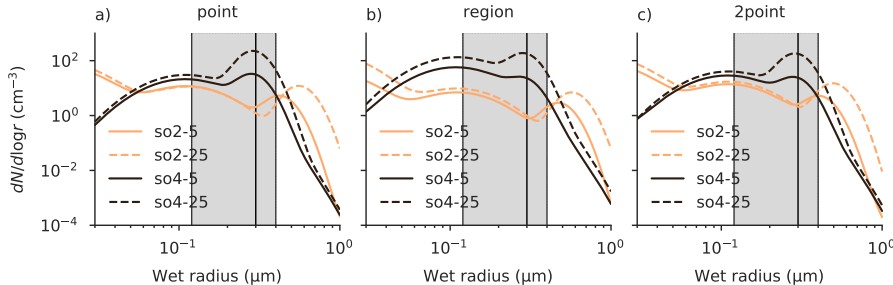

**Figure 10.** Global mean aerosol size distributions focusing on $AM-SO_4$ and coarse mode sulfate ($CM-SO_4$) particles at $62\,hPa$, which is the central level of the injection layer, for the ECHAM simulations of the point injections (a), the region injections (b), and the 2point injections (c). The grey bar marks the size range in which the backscattering efficiency of an aerosol particle with a given wet radius is at least $70\,\%$ (i.e. $0.12\,\mu m - 0.40\,\mu m$) of its maximum value, which is achieved for aerosols with a wet radius of $0.30\,\mu m$ and marked by a thick solid black line (Dykema et al., 2016).

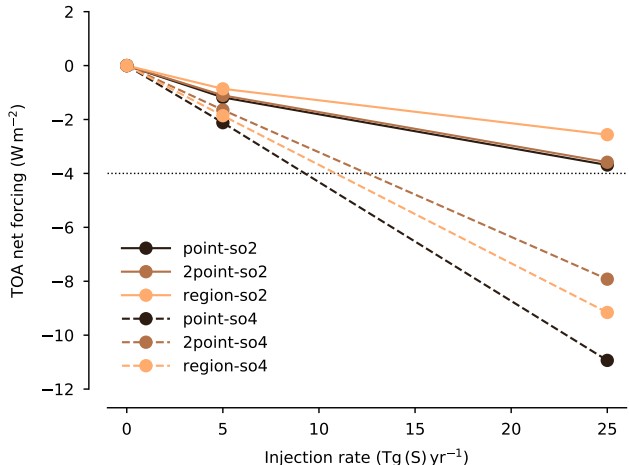

**Figure 11.** Global mean TOA all-sky net RF exerted by artificial sulfate aerosols as a function of injection rate. The dotted black line marks a RF of $-4\,W\,m^{-2}$.

the accompanied stronger warming of the lower tropical stratosphere relative to the mid-latitude one results in a stronger modification of $\overline{T}_{yy}$ (Fig. 9 a). However, the difference of the $\overline{T}_{yy}$ anomaly between both injection species is statistically significant at the $95\,\%$ level in a student's $t$-test only for the point injections. This causes a stronger QBO westerly phase for
an injection of $AM-SO_4$ compared to an injection of $SO_2$ as indicated by the anomalies of $\overline{u}$ (Fig. 9 b). The difference of the $\overline{u}$ anomaly between both injection species is statistically significant at the $95\,\%$ level in a student's $t$-test for both injection strategies, the point and the region injections. For the 2point injections of $25\,Tg(S)\,yr^{-1}$, an injection of $AM-SO_4$ instead of $SO_2$ causes the opposite effect as it slightly weakens the positive anomaly of $\overline{T}_{yy}$ and $\overline{u}$ within the tropics (Fig. 9).

The reason for the higher sulfate burden obtained for an injection of $AM-SO_4$ compared to an injection of $SO_2$ are differences in microphysical processes. Due to weaker coagulation and condensation, the sulfate particles stay on average smaller for an injection of $AM-SO_4$ than for an injection of $SO_2$ for all tested injection scenarios (Fig. 10). This reduces their sedimentation and enhances their stratospheric lifetime, which explains the larger sulfate burden. Additionally, smaller sulfate particles have a higher backscattering efficiency (Fig. 10). Therefore, the RF efficiency (RF per injected amount of sulfur) is also significantly higher for an injection of $AM-SO_4$ than for an injection of $SO_2$ (Fig. 11). The required injection rate to achieve a given RF is consequently clearly smaller for an injection of $AM-SO_4$ compared to an injection of $SO_2$. For example, to counteract a RF of $4.0\,\mathrm{W\,m^{-2}}$ as proposed in the GeoMIP6 experiment *G6sulfur* (Kravitz et al., 2015), an injection of $SO_2$ would require injection rates of more than $25\,\mathrm{Tg(S)\,yr^{-1}}$, while an injection rate of about $10\,\mathrm{Tg(S)\,yr^{-1}}$ to $12.5\,\mathrm{Tg(S)\,yr^{-1}}$ would be sufficient for an injection of $AM-SO_4$, depending on the injection strategy (Fig 11). The higher RF efficiency of an injection of $H_2SO_4$ should therefore be considered when comparing the QBO response between both tested injection species.

## 4 Comparison between ECHAM and CESM

Both models simulate a reasonable QBO in the control simulation (Fig. 12 a,b). With roughly 32 months the simulated QBO period of ECHAM is slightly longer than the one simulated in CESM, which is approximately 27 months. Both compare well to the observed period of 28 months on average (Naujokat, 1986). The simulated QBO winds, especially the QBO westerlies, are substantially stronger in ECHAM than in CESM at altitudes above $40\,\mathrm{hPa}$. Accordingly, the QBO easterly phases are longer and relatively stronger in CESM at altitudes below $30\,\mathrm{hPa}$. These general differences of the simulated QBO have to be considered when comparing the QBO response to different SAM scenarios in both models.

In the following two Sections, the QBO response to the 2point and region injections will be compared for ECHAM and CESM based on the injection of $AM-SO_4$ only. For an injection of $SO_2$ instead of $AM-SO_4$, the observed characteristics of the QBO remain basically the same in both models (See Sec. 3.3) and corresponding plots for an injection of $SO_2$ can be found in the supplementary material. A comparison of the QBO response to the point injection strategy is not possible, since the point injection is no part of the *accumH2SO4* experiment protocol and was, therefore, not performed by CESM.

### 4.1 2point injection strategy

For the 2point injections of $AM-SO_4$ simulated in ECHAM, the periodicity and strength of the QBO are basically not modified for the tested injection rates of $5\,\mathrm{Tg(S)\,yr^{-1}}$ and $25\,\mathrm{Tg(S)\,yr^{-1}}$ (Fig. 12 c,e). However, for an injection rate of $25\,\mathrm{Tg(S)\,yr^{-1}}$ a slight easterly anomaly of up to $-3\,\mathrm{m\,s^{-1}}$ has been noticed at approximately $40\,\mathrm{hPa}$ and $5°\,\mathrm{S}$ (Fig. 13 c), which is at the edge of extreme natural variability based on a student's $t$-test. In CESM, the QBO is also not modified much relative to the control simulation for the 2point injections with an injection rate of $5\,\mathrm{Tg(S)\,yr^{-1}}$ (Fig. 12 d). For an injection rate of $25\,\mathrm{Tg(S)\,yr^{-1}}$, the QBO basically maintains its oscillating behaviour in CESM as well (Fig. 12 f), but with clearly stronger easterlies and weaker westerlies at altitudes below $20\,\mathrm{hPa}$ compared to the control simulation.

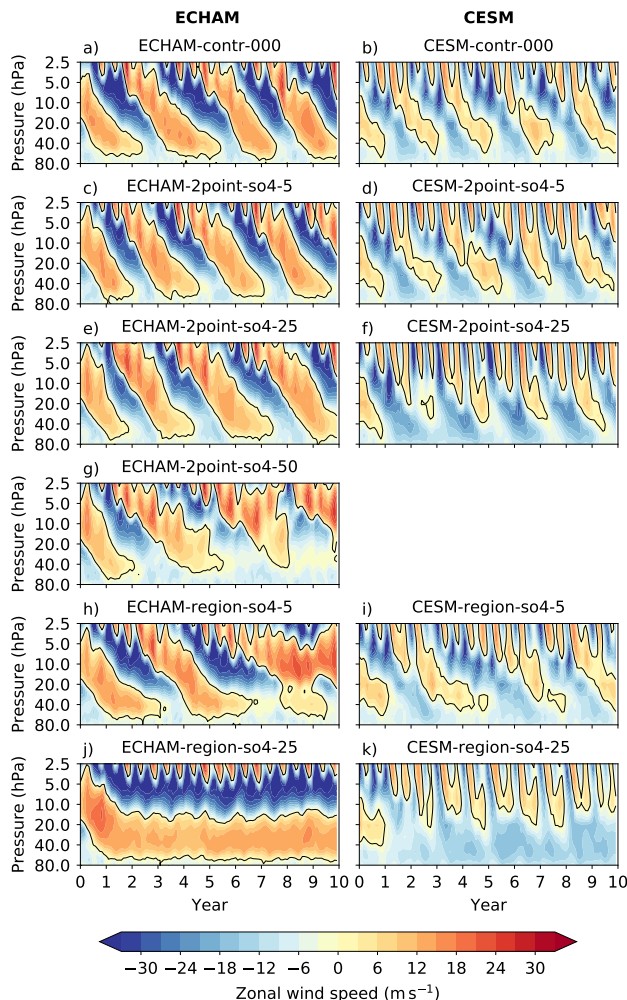

**Figure 12.** Time-height cross sections of the $5°$ N to $5°$ S mean zonal wind in the stratosphere over the simulation period of ten years for the $AM-SO_4$ injection scenarios in ECHAM (left column) and CESM (right column). The 1$^{st}$ row shows the control simulation, the 2$^{nd}$ row shows the 2point injections of $5\,Tg(S)\,yr^{-1}$, the 3$^{rd}$ row shows the 2-point injections of $25\,Tg(S)\,yr^{-1}$, the 5$^{th}$ row shows the region injections of $5\,Tg(S)\,yr^{-1}$, and the 6$^{th}$ row shows the region injections of $25\,Tg(S)\,yr^{-1}$. The 2point injection of $50\,Tg(S)\,yr^{-1}$ was only performed with ECHAM and is shown in the 4$^{th}$ row. The solid black line marks a tropical mean zonal wind speed of $0\,m\,s^{-1}$.

Nevertheless, the QBO responds qualitatively similar to a 2point injection of $AM-SO_4$ in both models, which is clearly shown by Figures 12 c-f. The spatial structure of the $\overline{u}$ anomalies, indicated by the contour lines in Figure 13, does in general agree for both models and both tested injection rates (Fig. 13 a-d). For an injection rate of $25\,Tg(S)\,yr^{-1}$, it shows a lower stratospheric easterly anomaly centered at approximately $40\,hPa$ and $5°$ S and an upper stratospheric westerly anomaly, which further extends into the northern hemisphere. The anomaly of $\overline{T}_y$ also shows basically the same spatial structure in both

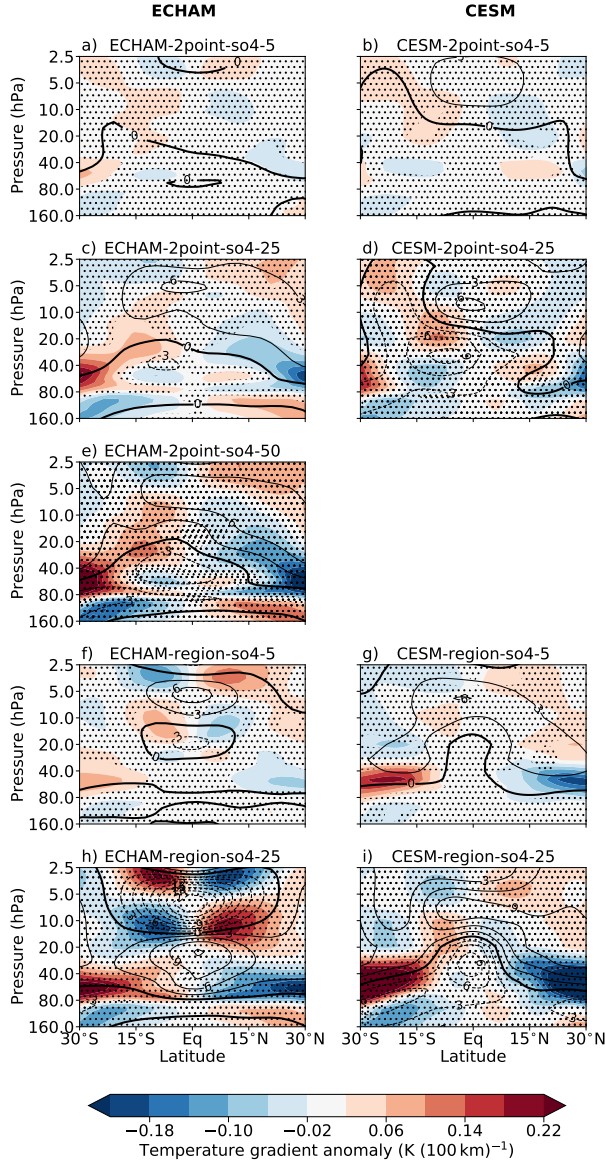

**Figure 13.** Latitude-height cross sections of the anomaly of the meridional zonal mean temperature gradient $\overline{T}_y$ for the $AM-SO_4$ injection scenarios in ECHAM (left column) and in CESM (right column). Stippling indicates areas where anomalies are not significant at the 95 % level in a student's $t$-test. Black contour lines indicate the anomaly of the zonal mean zonal wind speed $\overline{u}$ in intervals of $3\,\mathrm{m\,s^{-1}}$, with the thick black line denoting $\overline{u}{=}0\,\mathrm{m\,s^{-1}}$. Solid lines denote a westerly anomaly, while dashed lines denote an easterly anomaly. The 1st row shows the 2point injections of $5\,\mathrm{Tg(S)\,yr^{-1}}$, the 2nd row shows the 2-point injections of $25\,\mathrm{Tg(S)\,yr^{-1}}$, the 4th row shows the region injections of $5\,\mathrm{Tg(S)\,yr^{-1}}$, and the 5th row shows the region injections of $25\,\mathrm{Tg(S)\,yr^{-1}}$. The 2point injection of $50\,\mathrm{Tg(S)\,yr^{-1}}$ was only performed with ECHAM and is shown in the 3rd row.

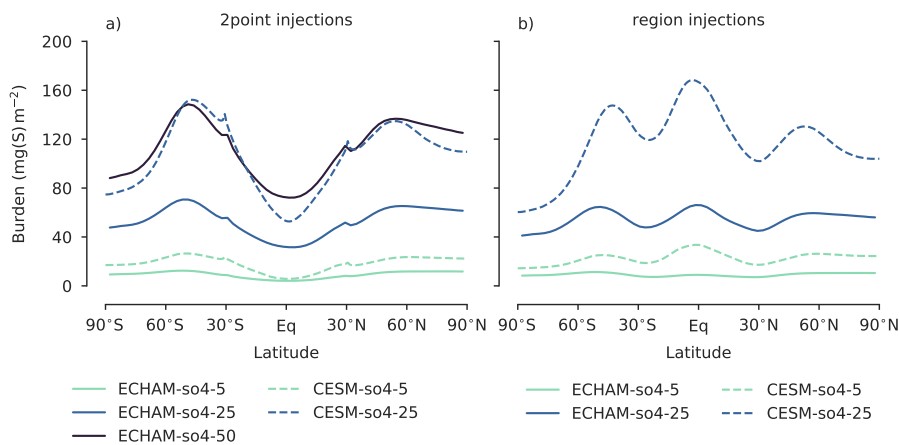

**Figure 14.** Zonal mean artificial sulfate burden for the 2point injections (a) and the region injections (b) of $\mathrm{AM-SO_4}$ as a function of latitude. Solid lines indicate the ECHAM simulations, dashed lines indicate the CESM simulations. The artificial sulfate burden can be used as a very basic proxy for the amount of heating due to absorption of OTLR and NIRR.

models as the usually positive poleward $\overline{T}_\mathrm{y}$ between approximately $80\,\mathrm{hPa}$ and $40\,\mathrm{hPa}$ strengthens statistically significantly. For a given injection rate, this strengthening – located approximately between $80\,\mathrm{hPa}$ and $40\,\mathrm{hPa}$ and between $20°\,\mathrm{S}$ and $20°\,\mathrm{N}$ – is clearly stronger and vertically more extended in CESM (Fig. 13 c,d), which explains the stronger easterly anomaly observed in CESM as a consequence of thermal wind balance (Eq. 1).

     Figure 14 a demonstrates the reason for the more significant strengthening of the poleward $\overline{T}_\mathrm{y}$ in CESM. In accordance with
Niemeier et al. (2020), the sulfate burden for a given injection rate is substantially larger in CESM than in ECHAM, which is predominantly due to a stronger $\overline{w}^*$. Given the characteristic meridional distribution of sulfate particles for a 2point injection, this results in a higher sulfate burden in the subtropical stratosphere relative to the tropical one in CESM (Fig. 14 a). This is the reason for the stronger modification of $\overline{T}_\mathrm{y}$ for a given injection rate in CESM compared to ECHAM.

     Following Niemeier et al. (2020), we therefore performed an ECHAM simulation of a 2point injection which results in
approximately the same global mean sulfate burden and the same meridional distribution of sulfate particles like the CESM simulation of a 2point injection with an injection rate of $25\,\mathrm{Tg(S)\,yr^{-1}}$. This is the case for an injection of $50\,\mathrm{Tg(S)\,yr^{-1}}$ in ECHAM (Fig. 14 a). As visible in Figure 12 g, the QBO easterly phases are substantially prolonged also in ECHAM for injections with an injection rate of $50\,\mathrm{Tg(S)\,yr^{-1}}$. In this simulation, the QBO westerlies in the lower stratosphere at altitudes below $20\,\mathrm{hPa}$ are clearly weaker than in the control simulation (Fig. 12 g). Overall, the spatio-temporal structure of the QBO
jets agrees reasonably well between the CESM simulation with an injection rate of $25\,\mathrm{Tg(S)\,yr^{-1}}$ and the ECHAM simulation of $50\,\mathrm{Tg(S)\,yr^{-1}}$, given the general differences of the simulated QBO of both models. Additionally, also the anomalies of $\overline{T}_\mathrm{y}$ and $\overline{u}$ agree reasonably well with each other (Fig. 13 d,e). This indicates that the QBO does in principle respond similarly to a 2point injection of $\mathrm{AM-SO_4}$ in both models, but that this response is more sensitive to an increase of the injection rate in

CESM than in ECHAM, which is in agreement with Niemeier et al. (2020). Nevertheless, the QBO response is still stronger in
CESM than in ECHAM and the reason for this has not been conclusively determined.

## 4.2 Region injection strategy

In contrast to the 2point injections, the response of the QBO to a region injection of $AM-SO_4$ is fundamentally different in
ECHAM and CESM. For an injection rate of $5\,\mathrm{Tg(S)\,yr^{-1}}$, the QBO slows down in both models with lower stratospheric
winds being predominantly westerly in ECHAM, while being more easterly in CESM (Fig. 12 h, i). For an injection rate
of $25\,\mathrm{Tg(S)\,yr^{-1}}$, the lower-stratospheric QBO is locked in a permanent westerly phase in ECHAM, while it is locked in a
permanent easterly phase in CESM (Fig. 12 j, k). Accordingly, in ECHAM $\overline{u}$ has a westerly anomaly of up to +12 $\mathrm{m\,s^{-1}}$ at the
equator at a pressure level of approximately $25\,\mathrm{hPa}$, while in CESM it has an easterly anomaly of more than -10 $\mathrm{m\,s^{-1}}$ at the
same location (Fig. 13 h,i).

For ECHAM, the results are explained by the weakening of the usually positive poleward $\overline{T}_y$ due to the aerosol-induced
warming of the lower tropical stratosphere, which induces additional westerly shear within the injection layer centered at
$60\,\mathrm{hPa}$ (see Sec. 3.1.3). Figure 13 h shows that the anomalies of $\overline{T}_y$ reach far into the tropics at altitudes between $20\,\mathrm{hPa}$
and $80\,\mathrm{hPa}$ for the ECHAM simulation of region-so4-25. In contrast, the anomalies within the same height range are only
weak between $10°\,\mathrm{S}$ and $10°\,\mathrm{N}$ in the corresponding CESM simulation (Fig. 13 i). They even slightly change sign locally. This
indicates that the warming of the lower tropical stratosphere relative to the mid-latitude one is clearly weaker in CESM than in
ECHAM. The resulting aerosol-induced temperature anomaly is meridionally more uniform in CESM, which corresponds to
the observed easterly anomalies of $\overline{u}$ in CESM.

However, Figure 14 b shows that the observed differences in $\overline{T}_y$ between both models cannot be explained by differences in
the meridional distribution of sulfate. For both models, the meridional distribution of sulfate basically exhibits the same shape
with a distinct equatorial peak and two additional local maxima located approximately at $50°\,\mathrm{S}$ and $50°\,\mathrm{N}$. Based on this, we
would have expected basically the same QBO response for both models.

## 4.3 Impact of ozone depletion on the QBO response

We assume that the significant difference in the QBO response to a region injection of $AM-SO_4$ between CESM and ECHAM
is explained at least partly by the interactive treatment of ozone in CESM. Figure 15 shows that in the CESM simulations,
the artificial injections of $AM-SO_4$ lead to a strong depletion of ozone at altitudes between $20\,\mathrm{hPa}$ and $40\,\mathrm{hPa}$. Thereby,
the strength and the location of the negative ozone anomalies closely correspond to the spatial distribution of the sulfate
aerosols as represented by the sulfate mass mixing ratio. For the region injections, this implies that the aerosol-induced lower
stratospheric heating by the absorption of OTLR and NIRR is at least partly compensated by a reduction of SW heating due to
ozone depletion, especially within the tropics in between $15°\,\mathrm{S}$ and $15°\,\mathrm{N}$. For example, the region injection of $25\,\mathrm{Tg(S)\,yr^{-1}}$
results in an ozone depletion of more than -1.5 $\mathrm{ppm}$ at the equator at an altitude of approximately $25\,\mathrm{hPa}$ compared to the
control simulation (Fig. 15 d). This corresponds to a change of about -30 %. Consequently, also the ozone-related SW heating
at this altitude would be reduced by approximately 30 %. We argue that for the region injections, this partial compensation

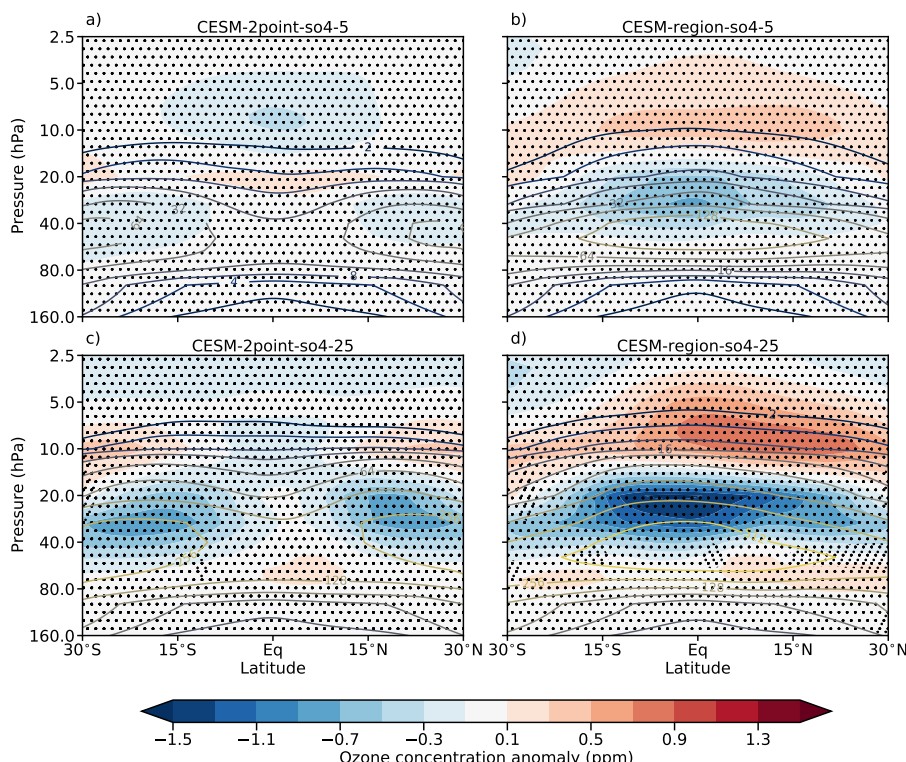

**Figure 15.** Latitude-height cross section of the ozone concentration anomaly for the simulations of an $AM-SO_4$ injection in CESM. Stippling indicates regions where anomalies are not significant at the 95 % level in a student's *t*-test. Contour lines indicate the zonal mean sulfate mass mixing ratio $\overline{m}_{SO4}$ in µg kg$^{-1}$. The contour interval is logarithmic starting at $2\,µg\,kg^{-1}$. The sulfate mass mixing ratio can be reasonably used as a proxy for the heating rate due to absorption of OTLR and NIRR.

of aerosol-induced heating by ozone-induced cooling is likely to contribute significantly to the observed different anomalies of $\overline{T}_y$ in the lower tropical stratosphere between $40\,hPa$ and $80\,hPa$ in CESM compared to ECHAM (Fig. 13 f–i). Thereby, the aerosol-induced changes in the ozone concentration help to prevent the QBO from being locked in a permanent lower-
stratospheric westerly phase like in ECHAM, which has no interactive ozone chemistry. Our theory is in agreement with ECHAM simulations of an equatorial point injection of $SO_2$ using a prescribed ozone field, which was interactively calculated in corresponding CESM simulations. These ECHAM simulations resulted in a weaker westerly anomaly of the QBO winds than the ECHAM simulations of this study (Niemeier, pers. communication).

However, based on our analysis we cannot fully explain why the QBO is locked in a strong permanent easterly phase
in CESM. The lower stratospheric ozone depletion as well as the upper stratospheric ozone increase alone may only partly account for this substantial difference between both of our models. Most likely, differences in the SAM-induced changes of the resolved and parameterized wave forcing of the QBO may explain its different response to SAM in both models. Additionally,

differences in the GW parameterization of both models itself are likely to account to the observed differences, as they are tuned to represent the QBO realistically in the current climate, but may react very differently to an external forcing like artificial sulfate aerosols.

For the 2point injections, changes of stratospheric ozone levels are mostly located outside the equatorial region (Fig. 15 a,c). Therefore, tropical SW heating may not be altered significantly, which explains why for the 2point injections the response of the QBO as an equatorial system was found to be in principle the same for both models.

### 4.4 Characteristics of ozone depletion in CESM

The ozone changes observed in the CESM simulations are consistent with previous simulations with the older version of the same model (CESM1(WACCM), see for instance Tilmes et al. (2017) and Richter et al. (2017)) and appear to be mostly independent from the type of injection, except for slight differences in the overall burden already discussed. The 2point simulations result in little changes in ozone concentration near the equator, due to lower $SO_4$ concentrations, while a decrease is observed in the subtropics and midlatitudes (Fig. 15 a,c). This ozone decrease is mostly driven by the increase in the surface area density (SAD) produced by the aerosols, that enhances the ozone destruction by halogens (such as HOCl) in the heterogeneous reaction (at $T < 200\,\mathrm{K}$)

$$ClONO_2 + H_2O \rightarrow HOCl + HNO_3 \tag{R1}$$

where $ClONO_2$ is one of the main stratospheric reservoir of inorganic chlorine (see Seinfeld and Pandis, 1998, Sec. 5.6). In the region simulations, on the other hand, the ozone changes result to be more similar to those observed in previous studies for equatorial point injections (Fig. 15 b,d). This is due to the higher $SO_4$ concentration in the tropical lower stratosphere, that also tends to extend in the middle stratosphere due to stronger upwelling. In this case, while the ozone reduction in the lower stratosphere can be attributed to the same heterogeneous chemistry mechanism described above, the ozone increase in the middle stratosphere can be explained by the predominance of the $NO_x$ cycle for the ozone budget at those altitude. The enhanced SAD results in a reduction in reacting nitrogen ($N_2O_5$, that is a catalyst for ozone destruction) due to the heterogeneous reaction

$$N_2O_5 + H_2O \rightarrow 2HNO_3 \tag{R2}$$

that, in turn, reduces the $NO_x$ driven ozone loss cycle (Visioni et al., 2017b). The two effects combine resulting, however, in no changes in the tropical stratospheric ozone column. At higher latitudes the similar $SO_4$ distributions result in identical ozone changes in the ozone column. For more details on stratospheric ozone depletion we refer to Tilmes et al. (2018).

### 5 Summary and Discussion

Within this study, we performed several simulations with the GCMs ECHAM and CESM to comprehensively compare the response of the QBO to different SAM setups with regard to the injection strategy, the injection rate, and the injection species.

Thereby, we aimed at a deeper investigation of the reasons for structural differences in the QBO response to different SAM setups. We identified the following key characteristics of the QBO response to SAM:

- The QBO response to SAM does fundamentally depend on the injection strategy. The injection rate and species rather act to scale the strength of this response.

- We clearly identified the meridional structure of the aerosol-induced temperature anomaly within the lower tropical stratosphere instead of its absolute strength as the key parameter explaining the observed different responses of the QBO to our different injection setups.

- For the equatorial point and for the region injections the aerosol warming peaks more or less sharply at the equator, causing a weakening of the poleward $\overline{T}_y$ in the lower tropical stratosphere. This generates a westerly wind anomaly and eventually forces the QBO into a permanent westerly phase.

- In contrast, $\overline{T}_y$ is basically not modified in the tropics for the 2point injections due to a meridionally nearly uniform warming. Therefore, the QBO remains approximately in its natural state.

Obviously, linking the QBO response to artificial sulfur injections to the meridional shape of the aerosol-induced temperature anomaly offers the possibility to explain the fundamentally different response of the QBO to all of our three injection strategies simulated with ECHAM in a stringent manner. This is a clear advancement compared to earlier studies, for example Niemeier and Schmidt (2017) or Tilmes et al. (2018), who did not adequately discussed differences in the QBO response between different injection strategies. The dependency of the QBO modification on the meridional structure of the lower stratospheric temperature anomaly via thermal wind balance may also be helpful to explain the significant acceleration of the QBO found by Richter et al. (2017) for extratropical single-point injections at $15°$ S, $15°$ N, $30°$ S, and $30°$ N, of which the causes were not finally determined.

Therewith, our results indicate that the modification of thermal wind balance in the lower tropical stratosphere between $40\,\mathrm{hPa}$ and $80\,\mathrm{hPa}$ is a simple but sufficient framework to explain the simulated differences in the QBO modification for our three tested injection strategies as it directly links the zonal wind anomaly to the $\overline{T}_y$ anomaly. Furthermore, we have shown that the different QBO responses to the different injection strategies correspond to the observed differences in the acceleration of the residual tropical upwelling in ECHAM. For the point and the region injections, the residual tropical upwelling increases statistically significantly, while it does not increase for the 2point injections, which corresponds to the weak modification of the QBO observed for the 2point injections in ECHAM. However, the reason for these differences is again the degree of tropical confinement of the aerosol-induced warming of the lower stratosphere because the modification of the residual circulation does ultimately also depend on the meridional gradient of the aerosol-induced temperature anomaly. This shows that using thermal wind balance as the overriding concept to explain the observed differences in the QBO response is reasonable.

An increase of the injection rate from $5\,\mathrm{Tg(S)\,yr^{-1}}$ to $25\,\mathrm{Tg(S)\,yr^{-1}}$ as well as an injection of $\mathrm{AM-SO_4}$ instead of $\mathrm{SO_2}$ act to strengthen the specific QBO response of all three injection strategies. This has been shown to be a consequence of the stronger warming of the tropical lower stratosphere relative to the subtropical one for the point and region injections, which

results in a stronger westerly anomaly. For the 2point injection, the increase of the injection rate causes the opposite effect as it weakens the warming of the tropical lower stratosphere relative to the subtropical one, which causes a stronger easterly anomaly. This is a clear advancement compared to earlier studies since the impact of an increasing injection rate has so far only been studied for equatorial point injections (Aquila et al., 2014; Niemeier and Schmidt, 2017). Additionally, our study is the first one explicitly investigating the QBO response to an injection of $H_2SO_4$, modelled as an injection of $AM-SO_4$. For an injection of $AM-SO_4$, we found the sulfate particles to stay on average clearly smaller than for a corresponding injection of $SO_2$, which we ultimately identified as the root cause for the observed stronger QBO response for an $AM-SO_4$ injection. Details on the aerosol microphysical background of an injection of $H_2SO_4$ can be found for example in Pierce et al. (2010), Benduhn et al. (2016), or Vattioni et al. (2019).

Compared to ECHAM, we found the QBO to be much more sensitive to artificial sulfur injections in CESM for the 2point and region injection. Niemeier et al. (2020) found basically the same for equatorial point injections, which they attributed to the significantly higher sulfate burden simulated in CESM due to an up to $70\%$ stronger $\overline{\omega}^*$ at the altitudes of the injection layer in CESM. Besides its in general higher sensitivity, we further found that the QBO response to artificial sulfur injections is basically the same in both models for the 2point injections, but fundamentally different for the region injections. For the region injection with an injection rate of $25\,\mathrm{Tg(S)\,yr^{-1}}$, the QBO is locked in a persistent westerly phase in ECHAM, but in a persistent easterly phase in CESM. We think that this QBO response in CESM partly is a result of local changes of the ozone concentration in the tropical stratosphere and its associated changes in SW heating. The reduction of the SW heating in the lower stratosphere due to ozone depletion partly compensates the LW heating by sulfate particles in this region, which results in a meridionally more uniform temperature anomaly and, accordingly, a weaker westerly anomaly above the injection layer following thermal wind balance. This important process can only be simulated with a complex aerosol chemistry module and, thus, not in ECHAM. For equatorial point injections, the role of ozone in determining the dynamical response to SAM has been already addressed by Richter et al. (2017). They found that for injections of $SO_2$ with an injection rate of $6\,\mathrm{Tg(S)\,yr^{-1}}$, the QBO is locked in a persistent westerly phase in their simulation with prescribed ozone values, while it maintains its oscillation – despite a significantly longer period of $\sim 42\,\mathrm{months}$ – in their simulation with an interactive ozone chemistry. These results also indicate that SAM-induced modifications of stratospheric ozone concentrations may act as an easterly force for the QBO, in accordance with the findings of our study. To assess the importance of ozone for the QBO response to SAM in more detail, corresponding CESM simulations of a region and a 2point injection with prescribed ozone values would be clearly desirable. They could give further evidence for a major role of ozone in altering the dynamic response to SAM, which ultimately may also feedback on the sulfate distribution and the aerosol RF itself. Consequently, the lack of an interactive ozone chemistry must be considered as a major shortcoming of ECHAM.

Nevertheless, changes in ozone and the associated SW heating alone cannot explain the substantial differences of the QBO response to a region injection between ECHAM and CESM. Besides differences in the representation of aerosol microphysics and in horizontal and vertical resolution, we think that differences in the GW parameterization most likely explain why the QBO responds so differently to a region injection. Schirber et al. (2014) exemplarily showed for the ECHAM6 model (Stevens et al., 2013) that the simulated GW forcing in a GCM is highly dependent on the chosen GW parameterization. Additionally,

it is likely that the response of the GW forcing to SAM in general and its differences among the tested injection scenarios are not well captured in our simulations, which introduces additional uncertainty. However, a detailed assessment of the GW parameterizations and the resulting GW forcing of the QBO for both models would have gone beyond the scope of our study. Also changes in other forcing terms of the QBO could not have been assessed since the CESM data was only available on a monthly mean basis, which prevented us from performing a TEM analysis for CESM. Overall, the simulation of the QBO response to artificial sulfur injections critically depends on multiple factors, which is further complicated by feedback processes of an altered QBO onto the sulfate distribution and associated dynamical changes (e.g. Niemeier and Schmidt, 2017; Visioni et al., 2017a). Therefore, unexpected consequences for the QBO due to SAM would be likely and more research is necessary to avoid unintended negative side effects of SAM. Furthermore, not a single solar geoengineering method would be able to reproduce a climate state similar to a natural one with the same global mean temperature. Consequently, we think that a substantial reduction of anthropogenic GHG emissions is still the only responsible way of preventing a drastic global warming.

*Code and data availability.* Primary data and scripts used in this analysis and other supplementary information that may be useful in reproducing the author's work are available under: https://cera-www.dkrz.de/WDCC/ui/cerasearch/entry?acronym=DKRZ_LTA_550_ds00005. Model results of MAECHAM5-HAM and corresponding metadata are available under: https://cera-www.dkrz.de/WDCC/ui/cerasearch/entry?acronym=DKRZ_LTA_550_ds00003. The CESM2(WACCM) model code can be obtained via https://github.com/ESCOMP/CESM. Model results of CESM2(WACCM) can be obtained by contacting DV.

*Author contributions.* HF performed the ECHAM simulations with strong support from UN. DV performed the CESM simulations. HF wrote the paper with contributions of DV on CESM description and results. All authors discussed the idea and results of this study.

*Competing interests.* We declare no competing interests.

*Acknowledgements.* HF wants to thank UN and Stefan Bühler (University of Hamburg, Germany) for enabling him to work on this very exciting Master's thesis project, for the excellent supervision, and for the possibility to publish parts of his results in this paper. We thank Yaga Richter (NCAR) for providing input data, Debra Weisenstein and David Keith (Harvard) for giving valuable input and tips on the simulation setup and for giving the impulse for this project, Marco Giorgetta (MPI-M) for giving very helpful comments on the first manuscript, and three anonymous reviewers for their helpful suggestions. ECHAM simulations have been performed on the computer of Deutsches Klimarechenzentrum (DKRZ). UN got support from German DFG-funded Research Unit VollImpact FOR2820 sub project TI344/2-1 and DFG-funded Priority Program 'Climate Engineering: Risks, Challenges, Opportunities?' (SPP 1689). The CESM project is supported primarily by the National Science Foundation. This work was supported by the National Center for Atmospheric Research, which is a major

facility sponsored by the National Science Foundation under Cooperative Agreement No. 1852977. Support for DV was provided by the Atkinson Center for a Sustainable Future at Cornell University and by the National Science Foundation through agreement CBET-1818759.

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
