# Peer review of "Differences in the QBO response to stratospheric aerosol modification depending on injection strategy and species"

_Atmospheric Chemistry and Physics, 2020_

## Referee Comment (RC1) · Anonymous Referee #1 · 7 Dec 2020

This paper documented changes in QBO in responses to stratospheric aerosol modification simulated in ECHAM and WACCM. The authors compared three injection locations, two injection rates and two injection species. It is found that the QBO is strongly disturbed when aerosols are injected at the equator or evenly over the tropics, but not when aerosols are injected at two subtropical points, and H2SO4 has a stronger effect on QBO than SO2. The two models simulated different responses in QBO for the region injection case, which is attributed to the ozone changes. The authors explained these difference by the thermal wind relationship and linked the wind shear to the temperature curvature. This paper provides a thorough discussion on how geo-engineering affects QBO. It is logically organized and clearly written. I do have one major concern

regarding the interpretation of the thermal wind relation. I recommend the publication of the paper after the authors address this comment.

Major comment:

The authors attributed the equatorial winds to the temperature curvature, and argued that aerosols drives temperature changes which then drives wind changes. There is no doubts that the thermal wind relationship is valid, but it is important to realize that the thermal wind relationship itself does not guarantee a causal relationship. The agreement between temperature curvature and wind shear shown in the paper is not sufficient to prove that temperature changes drive the winds. This is particularly the case for levels above ∼20 hPa, which are well above the aerosol layers so that the direct radiative effect from the aerosol should be quite weak. Yet, these upper level changes are key to the QBO changes in many cases. An example would be the ECHAM region-so4-5 vs region-so4-25 (Fig. 12 f vs h). These two simulations only differ in the magnitudes of the aerosol injection, but the changes in temperature gradient are opposite above 20 hPa, which suggested to me that the temperature changes there are not directly related to the aerosol's radiative effect. It may be helpful to plot the changes in the radiative heating rate from the aerosols (and ozone).

Minor comments:

Line 96-97: The radiative heating of the sulfate aerosols consists of long wave and near-infrared radiation, but aerosol properties are only passed to shortwave and near infrared radiation. Is the aerosols' long wave radiative effect ignored?

Line 118: These AMIP simulations cannot simulate the surface cooling by the sulfate layer as most of the surface temperature is constrained. But they can be used to estimate the forcing.

Line 207-210: T/Lˆ2 basically provides a scaling for the temperature curvature. So the equation (3) may still hold if choosing a correct "L".

Line 251: Fig. 2 does not show the comparison between the two injection rates.

Line 397-401: The authors attributed the ozone increase above the aerosol layers to the changes in NOx. I am not sure how NOx changes at 10 hPa or higher where there is almost no sulfate aerosols. It is also interesting to compare the CESM-region-so4-5 vs CESM-region-so4-25 (Fig. 14 b vs d). The aerosol layer is much deeper in the latter case, but the vertical structure of the ozone changes does not seem to vary much between the two. Also, does the temperature dependence of the reaction rates play any role here?

---

## Referee Comment (RC2) · Anonymous Referee #2 · 12 Dec 2020

The paper by Franke et al. examines how the QBO varies in response to the injection of stratospheric aerosols in a variety of different locations (single equatorial grid point, two subtropical gridpoints, and in a zonally symmetric sense over 30S-30N) using two GCMs. They find that the QBO response is qualitatively sensitive to the injection strategy with the two-point method yielding no QBO change and the other two methods yielding a QBO change. They also find that the QBO is quantitatively sensitive to the magnitude of the aerosol injection as well as to the injected aerosol type (be it SO2 or H2SO4). They finally demonstrate that ozone plays a role in why the injected aerosol type matters; interactive ozone chemistry is likely required in any future studies on such geoengineering procedures. Overall, they claim that arguments related to thermal wind

balance are sufficient to explain the changes and sensitivities that are found, although I do have doubts about how strong their causation arguments are using this inherently diagnostic relationship (see comments below).

I found the manuscript to be very interesting and well-written. My comments are mostly based on clarity and improving the presentation and although I have written a number, please take this as me being genuinely interested in the results. Hence, my suggestion is of minor revisions.

Minor and technical comments: Lines 31-32; Can you state explicitly here why this led to a prolonged westerly QBO phase? Did Labitzke (1994) provide reasoning? Presumably because the temperature anomaly leads to stronger upwelling below the westerlies that counteracts the downwelling that is ordinarily associated with the westerlies?

Line 44; 'further' –> 'also'

Line 54; Not sure what this sentence means. Please rephrase. Do you mean qualitatively similar but quantitatively different?

Line 75; delete 'one'

Line 85; which parameterisation? Can you be more specific here?

Line 124; Why is CESM not run using the single-point injection setup?

Table 1; it needs to be stated whether the total aerosol load injected in the, e.g., 25Tg experiments is the same across the different injection locations - i.e., is the 25Tg version of the region experiment spread over the entire zonal band so that the 25Tg point source run is inputting the same amount of aerosol? Otherwise I fear that one is not comparing apples to apples when one writes sentences such as on lines 150-151. A clearer example is whether the 2-point run has two point sources of 12.5Tg to match the 25Tg in the single-point source run. You also need to presumably take into account the grid box size in this calculation as a 25Tg single point run at the equator is not equivalent to a 25Tg run at 15N due to the smaller grid box size further poleward.

Lines 131-133; I think a bit of background as to why these values are prescribed would be useful. Are they typical of the amount of aerosol injected in a volcanic eruption? Or are they generally used in current literature?

Section 3.1; The word 'disruption' in the title seems like a bit of a misnomer as thermal wind balance holds. The injection that causes changes in temperature and wind are in thermal wind balance with one another.

Line 172; why is this experiment chosen to present herein? Is there a linear relationship between the experiments so that the temperature anomaly in the lower stratosphere for the so2-5 run is five times smaller than the presented so2-25 run? Also, change 'exemplary' to 'exemplarily' (also in other places in the manuscript).

Line 174; what does 'usually positive poleward' mean here? Fig 3a shows the temperature gradient anomaly in the experiments where the aerosol is injected. Do you mean the Ty anomalies associated with the QBO in the control run?

Lines 183-184; Does figure 2b really show a more uniform meridional heating profile?

Line 185; still in the lower stratosphere though so I think this is worth reiterating. Aloft, it is not true.

Lines 196-197; I do not see this slight intensification of Ty between 15S and 15N - if anything there is a dipole across the equator. Please clarify. Lines 170-201; I think you need to be careful about the causation argument here. Indeed, thermal wind balance holds even at the equator. But the wording is very much stating that the aerosol injection changes the temperature structure which then leads to wind changes. Thermal wind balance cannot explicitly tell us this. This is even more clear aloft where there are large Ty and Tyy anomalies where the aerosol injection does not reach and it is hence the causation is less clear.

Lines 203-206; This is true when comparing different injection locations, but not true if you compare different magnitudes injections at the same forcing locations (I also asked

before, is there a linearity in the temperature anomaly and wind anomaly in response to different magnitude injections? It appears that there might well be from figure 4 when comparing the 2xblack-line experiments). As I also stated, I am worried that comparing the different forcing locations is misleading if the overall injection magnitude is not the same.

Lines 207-210; I do not follow this. Can you be clearer here?

Lines 223-224; You only state this observation here in the runs, but it is not unexplained. It is not clear to me why the lower stratospheric anomalies lead to such strong w anomalies aloft compared to closer to the injection location. Please explain if it is known why.

Line 230; the static stability changes in the tropical stratosphere though, right? Is there also a change at higher latitudes that affects the planetary wave propagation in the way that Chen and Robinson (1992) espoused? My guess is that it is due to the change in winds at higher latitudes that changes in planetary waves occur.

Lines 237-238; How can you attribute the w>0 anomalies in fig 5 to the SMC as opposed to just a change in the BDC?

Line 247; Is there a threshold above which the injection acts to no longer drive an oscillating QBO but instead drives constant westerlies/easterlies? At some injection rate between 5 and 25, there must be a crossover, the knowledge of which would presumably be useful for policy planning.

Figure 6; The 5S-5N meridional average is not a good measure for Tyy given that Tyy looks pretty symmetric about the equator (fig 3) so that the two likely cancel. I suggest to split into 5S and 5N (which should likely be antisymmetric).

Figure 7; at what height is this line plot?

Lines 271-274; Is the difference between these different injection-species runs significant? It looks to be a very minor difference (also between 2nd and 3rd columns of fig

1).

Figure 10; the final sentence of caption is confusing given that one of the experiment lines is also dashed black. I also think that you need to perform experiments with either 10 or 15Tg rates so that you can more confidently draw the lines of best fit as you have here. This harks back again to the linearity point I made before.

Figure 11; It would be useful if you included the periodicity of each QBO run in each individual figure panel so that the reader can immediately read it off without having to find the associated text.

Figure 12; the thick black denoting u=0 is difficult to distinguish from the thin contours. Please thicken.

Lines 308-309; 'in principle' –> 'qualitatively' given that 12c-d have similar spatial u anomalies. However, figures 12a-b essentially show no difference at all so it is being generous to include them in this sentence.

Lines 313-314; In fig. 12c,d I do not agree that the Ty is stronger and more poleward spread in CESM. In fact, the opposite looks true with ECHAM having stronger Ty anomalies extending poleward. Please clarify.

Line 318; I do not follow why the smaller AM-SO4 sulfate particles can explain the difference in Ty given that, presumably, they are the same size in the two models. I may be misunderstanding something but please make clearer.

Line 330; 'on' –> 'with'

Line 338; This is true only for the lower-stratospheric QBO.

Line 357; correspond to the positive anomalies of the sulfate mass mixing ratio? i.e., there is an anticorrelation.

Lines 363-368; Again, this is assuming a chain of causation using the thermal wind balance. Given it's inherent diagnostic nature, it is not possible to assume with such

certainty this causation chain. It is advisable to be more speculative here.

Line 371; partial

Lines 369-382; Another way to check (that is more robust I think as it brings the two models closer to one another) would be to switch off the interactive ozone in CESM and rerun the two experiments corresponding to figures 11j,k. Then you could see if the two models have similar QBO winds (i.e., does the CESM relinquish its constant lower-stratospheric easterly phase?). I am not sure how difficult this is to do however - is there a simple namelist parameter that can turn off ozone in CESM.

Eqs R1-R2; For someone that isn't an atmospheric chemistry wonk such as myself, these equations don't mean too much unless they are properly introduced (a couple of the terms are obvious, but most are not!). Can you explain each of the terms so that everybody can understand?

Lines 419-426; Please see prior comments on thermal wind balance. These are very strongly worded claims using an equation that does not yield itself well to causation arguments.

Line 466; assess
* * *

---

## Referee Comment (RC3) · Anonymous Referee #3 · 21 Dec 2020

This article focuses on changes to the QBO resulting from injections of SO2 and H2SO4 into the stratosphere in two global models. Multiple experiments are carried out with each model. It is shown that the QBO does not respond in the same way in the two models and the authors attempt to address the reasons for this building on previous studies on this topic.

The paper is well organized, well written and of scientific value. I do have one major concern though about the interpretation of the results as described below that needs to be addressed before the paper can be published.

Major Concern:

[Figure]

1. The authors explain changes to the QBO in terms of changes to thermal wind balance. Although changes to thermal well balance are there, I am not convinced that they explain how the QBO changes. The QBO is a wave driven phenomenon and its period and amplitude arise from an interaction of large scale waves (resolved in the models), parameterized gravity waves, and vertical and meridional advection (meridional advection is the least important driver) – it is hence most likely that changes to these factors are of crucial importance to the QBO. Vertical advection is mentioned in this paper but I'm not sure why it's not addressed more carefully. Changes in vertical advection directly affect the downward propagation of QBO phases (or the lack of ) hence this should be given a larger consideration. Thermal wind balance indeed maybe a factor as it can change the propagation of waves that drive the QBO, but it's difficult to accept based on what is shown here, how a change in the mean state (and balance) explains changes to a tropical oscillation that's primarily wave driven.

Minor issues:

1. Line 303: "slight easterly anomaly up to -3 m/s' – this could be natural variability of the QBO, and not really a change

———————————————————————

---

## Author Comment (AC1) · 5 Feb 2021

**Answers to the reviewers on the ACPD paper (acp-2020-1104): "Differences in the QBO response to stratospheric aerosol modification depending on injection strategy and species"**

Henning Franke[1, 2], Ulrike Niemeier[1] and Daniele Visioni[3]

[1] Max Planck Institute for Meteorology, Bundesstr. 53, 20146 Hamburg, Germany

[2] International Max Planck Research School on Earth System Modelling, Bundesstr. 53, 20146 Hamburg

[3] Sibley School for Mechanical and Aerospace Engineering, Cornell University, Ithaca, NY, USA

We thank all three anonymous reviewers for their valuable feedback. We considered their recommendations carefully and made some major changes in the manuscript. Please find below our answers to their comments. Their comment/question is in "**bold**" letters. Our answer is *italic* letters. Changes in the manuscript are blue and parts of the manuscript which remained unchanged are in "normal" black letters.

**General remarks concerning all reviews**

*In order to address the major concerns brought up by reviewers 1 and 3, we changed the structure of the manuscript in Section 3. In the preprint, Section 3 was structured as follows:*

Section 3.1 Dynamic mechanisms of QBO modification: Disruption of thermal wind balance

Section 3.2 Dynamic mechanisms of QBO modification: Modification of the residual circulation

Section 3.3 Impact of injection rate

Section 3.4 Impact of injection species

*In the revised manuscript, Section 3 is structured as follows:*

Section 3.1 Dynamic mechanisms of QBO modification

      Section 3.1.1 Aerosol-induced heating of the lower stratosphere

      Section 3.1.2 Modification of the residual circulation

      Section 3.1.3 Modification of thermal wind balance

Section 3.2 Impact of injection rate

Section 3.3 Impact of injection species

*We put most of the text of Section 3.1 in the preprint into Section 3.1.3 in the revised manuscript and most of the text of Section 3.2 in the preprint into Section 3.1.2 in the revised manuscript, but also added some further text into those sections. Consequently, the content of these Sections has changed considerably. Due to the restructuring of Section 3 also the order of some Figures changed and, furthermore, we added an additional Figure to the manuscript. Therefore, Figure numbers have changed throughout the whole manuscript compared to the preprint. In our answers to the reviewers, Figure numbers refer to those in the revised manuscript and the corresponding Figure numbers in the preprint are given in brackets.*

*Besides, we added an additional Section 3.1.1 "Aerosol-induced heating of the lower stratosphere", which was not included in the preprint in order to highlight the importance of the aerosol-heating as the initial perturbation.*

*Details on why we chose to employ specific changes can be found in our answers to the individual*

*reviewers. Please find the restructured and supplemented Section 3.1 of the revised manuscript below:*

[revised manuscript text omitted]

25hPa and 8hPa in region-so2-25), this increase of w* is superimposed by changes of the secondary meridional circulation (SMC) of the modified QBO itself. During a permanent QBO westerly phase, the SMC would also be permanently locked in its corresponding "westerly" phase, which acts to increase w* within the tropical stratosphere (Plumb and Bell, 1982). Our experiments indicate that a large fraction of the increase of w* in the upper tropical stratosphere (Fig. 5 a,b) (Fig. 3 a,b) can be attributed to this "indirect" acceleration via the SMC, especially in the upper stratosphere. For example, in the experiment point-so2-5, the tropical w* increased by up to 100% in the upper stratosphere (not shown). In contrast, in ECHAM simulations with permanent lower stratospheric easterlies instead of a QBO, Niemeier et al. (2011) obtained an increase of the tropical w* of only 5 to 10% for an equatorial point injection of $SO_2$ with an injection rate of $4Tg(S)\ yr^{-1}$. Therefore, one has to be cautious when interpreting the strong positive anomalies of $-w^*u_z$ observed in the upper stratosphere in point-so2-25 and region-so2-25 as the primary cause for the disruption of the QBO since they are – at least partly – rather its consequence.

Within the TEM framework, the characteristics of the general acceleration of the BDC can be further directly linked to the tropical confinement of the aerosol-induced temperature anomaly, as shown by Dunkerton (1983) in a study on the effect of the 1963 eruption of Mt. Agung on the QBO. Following the equation of the residual mean meridional streamfunction (Plumb and Bell, 1982), only a tropically confined heating modifies the BDC, whereas a meridionally uniform heating has no effect. This explains why the residual tropical upwelling increases in point-so-25 and region-so2-25, but not in 2point-so2-25: While in point-so2-25 and region-so2-25 the aerosol heating has an equatorial peak and decreases rather sharply towards the extratropics, in 2point-so2-25 it is meridionally nearly uniform within the tropics (see Sec. 3.1.1). Consequently, it is ultimately the meridional shape of the aerosol-induced lower stratospheric temperature anomaly, or – simply spoken – the degree of tropical confinement of the artificial sulfate aerosols, what determines the QBO response to artificial sulfate injections.

**3.1.3 3.1 Dynamic mechanisms of QBO modification: Modification of thermal wind balance**

The dynamic mechanisms which cause the observed modification and breakdown of the QBO for an equatorial point injection of $SO_2$ have been investigated by Aquila et al. (2014). They assume that, besides an increase of the tropical upwelling, a modification of the thermal wind balance in the tropical stratosphere due to the aerosol-induced warming is the main reason for the modification of the QBO. Besides via an increase of the tropical upwelling in the rising branch of BDC, Aquila et al. (2014) also attributed the observed changes in the QBO to a modification of thermal wind balance.

Thermal wind balance is an atmospheric state equation, which links the zonal mean meridional temperature gradient $T_y$ to the zonal mean vertical wind shear $u_z$. It is defined as

$$u_z = -R(H\beta y)^{-1}T_y \qquad (1)$$

for an equatorial β-plane (Holton, 2004). Assuming equatorial symmetry of the zonal mean temperature field, one can set $T_y = 0\ K\ km^{-1}$ at the equator and apply the rule of L'Hospital (Holton, 2004):

$$u_z = -R(H\beta)^{-1}T_{yy}. \qquad (2)$$

Within Equation 1 and 2, R denotes the gas constant for dry air, H the scale height, and β the meridional gradient of the Coriolis parameter at the equator. $T_{yy}$ denotes the meridional curvature of the zonal mean temperature.

~~Our simulations with ECHAM clearly confirm the results of Aquila et al. (2014), showing that the modification of thermal wind balance is the main driver for changes of the QBO by equatorial point injections. In the following, this will be exemplary demonstrated for the experiment point-so2-25. Due to the equatorial injection location, the sulfate aerosols are strongly concentrated within the tropics, which leads to a strong warming centered at the equator (Fig. 2 a). This warming~~

However, despite QBO changes due to artificial sulfur injections are frequently interpreted as the consequence of an increased residual tropical upwelling and a modification of thermal wind balance, one can't see both as two separate processes. In contrast, the acceleration of the BDC discussed in Section 3.1.2 is rather the mechanism by which thermal wind balance is reestablished for the aerosol-induced lower stratospheric temperature anomaly (Holton, 2004). While the differences in the increase of w* directly explain the different QBO response to different injection strategies in first order (see Sec. 3.1.2), in this Section we show that the differences in the QBO response can be also linked directly to the meridional shape of the aerosol-induced lower stratospheric temperature anomaly via thermal wind balance.

As discussed in Section 3.1.1, an equatorial point injection results in a significant positive temperature anomaly centered at the equator (Fig. 2 b). In the climatological mean – without artificial sulfur injections and represented by contr-000 – the lower tropical stratosphere is much colder than the lower midlatitudinal stratosphere (Fig. 4), leading to a poleward $T_y$ that is positive. The aerosol-induced warming abates the  poleward $T_y$ (Fig. 5 a) within the lower tropical stratosphere between 40hPa and 80hPa  which is accompanied by a significant negative anomaly of $T_{yy}$ centered at the equator  (Fig. 5 b). Following Equation 2, a negative anomaly of $T_{yy}$ results in stronger westerly shear. Consequently, a strong westerly anomaly of the zonal mean zonal wind u is located on top of the injection layer in order to maintain thermal wind balance  (Fig. 5 b). This results in the observed constant westerly QBO phase (Fig. 1 b).

For region-so2-25, the QBO was also found to be locked in a permanent westerly phase, but the vertical extent as well as the strength of the westerlies is weaker than for point-so2-25, which is in agreement with the results of Niemeier and Schmidt (2017). The reason for the weaker westerlies is the meridionally more uniform temperature anomaly within the lower tropical stratosphere  (Fig. 2 d). Therefore, the strongest modifications of $T_y$ in the lower stratosphere are located poleward of approximately 20°N and 20°S, while its modification close to the equator is relatively small  (Fig. 5 c). Accordingly, also the negative anomaly of $T_{yy}$ and – following thermal wind balance (Eq. 2) – the induced anomaly of westerly shear  is weaker near the equator compared to point-so2-25  (Fig. 5 d). Consequently, the lower stratospheric westerly anomaly of u is weaker for region-so2-25 than for point-so2-25.

For 2point-so2-25, the QBO was not found to be modified significantly and it basically preserved its natural periodicity (Fig. 1 k). Due to the extratropical injection locations, the highest sulfate concentrations are located at approximately 20°N and 20°S and so is the associated heating  (Fig. 2 e). Therefore, the lower-stratospheric temperature anomaly is meridionally nearly uniform between approximately 15°N and 15°S  (Fig. 2 f). Consequently, $T_y$ as well as $T_{yy}$ are  only very weakly modified between 40hPa and 80hPa and between approximately 15°S and 15°N (Fig. 5 e,f) and these anomalies are hardly statistically significant. Following thermal wind balance, this  implies that zonal wind anomalies in the lower stratosphere are small as well (Fig. 5 e,f), which is in accordance with the QBO in principle remaining in its natural shape.

~~When looking in more detail, the lower stratospheric temperature anomaly even has a slightly convex shape between approximately 15°N and 15°S for the 2point injections (Fig. 2 c), which is in contrast to the point and region injection strategies. Therefore, the usually positive poleward $T_y$ in the lower stratosphere (40 hPa to 80 hPa) slightly intensifies between approximately 15°S and 15°N (Fig. 3 e). This leads to a slight positive anomaly of $T_{yy}$ centered at the equator and approximately 50 hPa (Fig. 3 f)~~

~~and is accompanied by an anomaly of easterly shear following Equation 2. Accordingly, u has a slight easterly anomaly above the injection layer in 2point-so2-25 (Fig. 3 bottom). The consequence is a reduction of the asymmetry between the westerly and easterly QBO phases resulting (Fig. 1 bottom). However, this slight speed up of the QBO phase is not significant based on our short simulation period.~~ As discussed in Section 3.1.1, the negative temperature anomalies above approximately 20hPa have been induced dynamically due to an increased tropical upwelling and negative temperature advection (see Aquila et al., 2014). While the corresponding anomalies of $T_y$ and $T_{yy}$ above 20hPa are of course still in thermal wind balance with the upper stratospheric wind field (Fig. 5), it is important to mention that this agreement does not imply causality in the way that these upper stratospheric temperature anomalies have caused the lower stratospheric westerly anomaly and QBO modification. Our  results clearly show that differences in the QBO response with respect to our three tested injection strategies are linked to differences in the meridional structure of the aerosol-induced temperature anomaly. Therefore, the absolute strength of the aerosol-induced lower stratospheric temperature anomaly does not permit a statement about the strength of the QBO modification when comparing different injection strategies. For instance, the tropical (i.e. 5°N to 5°S) mean temperature anomaly within the injection layer is more than twice as high in 2point-so2-25 than in point-so2-5  (Fig. 6). However, the QBO is heavily perturbed in point-so2-5, while for 2point-so2-25 it remains nearly unchanged (Fig. 1 d,k). This comparison shows that within

$$u_z \sim R(H\beta)^{-1}L^{-2}T , \quad (3)$$

which is often used as an approximation of thermal wind balance for QBO variations centered at the equator (Baldwin et al., 2001),  the scaling factor L depends on the injection strategy. Therefore, Equation 3 cannot be used when comparing the QBO response to different injection strategies.

Since it is the degree of tropical confinement of the artificial sulfate aerosols what is ultimately decisive for the observed QBO response also when explaining the observed QBO changes solely as the consequence of an increased residual tropical upwelling, we will use thermal wind balance in our argumentation throughout this study as it directly links the observed QBO changes to the observed aerosol-induced temperature anomalies.

*Following this chain of argumentation, we changed some text passages of Section 5 (Summary and Discussion) considerably in order to reflect our changes in Section 3.1 and formulated them more defensive by removing thermal wind balance as the key concept at some points. Please find those changes below:*

**5. Summary and Discussion**

(…)

– For the equatorial point and for the region injections the  aerosol warming peaks more or less sharply at the equator, causing a weakening of the poleward $T_y$ in the lower tropical stratosphere. This generates  a westerly wind anomaly and eventually forces the QBO into a permanent westerly phase.

– In contrast, $T_y$ is basically not modified in the tropics for the 2point injections due to a meridionally nearly uniform warming. Therefore, the QBO remains approximately in its natural state.

 Obviously, linking the QBO response to artificial sulfur injections to the meridional shape of the aerosol-induced temperature anomaly offers the possibility to explain the fundamentally different response of the QBO to all of our three injection strategies simulated with ECHAM in a stringent manner. This is a clear advancement compared to earlier studies, for example Niemeier and Schmidt (2017) or Tilmes et al. (2018), who did not adequately discussed differences in the QBO response between different injection strategies. The dependency of the QBO modification on the meridional structure of the lower stratospheric temperature anomaly via thermal wind balance may also be helpful to explain the significant acceleration of the QBO found by Richter et al. (2017) for extratropical single-point injections at 15°S, 15°N, 30°S, and 30°N, of which the causes were not finally determined.

Therewith, our results indicate that the modification of thermal wind balance in the lower tropical stratosphere between 40hPa and 80hPa is a simple but sufficient framework to explain the simulated differences in the QBO modification for our three tested injection strategies as it directly links the zonal

wind anomaly to the $T_y$ anomaly. Furthermore, we have shown that the different QBO responses to the different injection strategies correspond to the observed differences in the acceleration of the residual tropical upwelling in ECHAM. For the point and the region injections, the residual tropical upwelling increases statistically significantly, while it does not increase for the 2point injections, which corresponds to the weak modification of the QBO observed for the 2point injections in ECHAM. ~~However, a comparison to ECHAM simulations with a permanent QBO easterly phase (Niemeier et al., 2011) indicate that a substantial part of the tropical upwelling anomaly can be attributed to the SMC of the modified QBO itself. The impact of the acceleration of the BDC as a whole, which is caused by an increased extratropical wave forcing as described in Tilmes et al. (2018) and Bittner et al. (2016), onto the QBO response may be rather small. Therefore, the increase of -ω*u and the associated slowdown of the QBO downward propagation maybe seen partly as a self-maintaining process of the perturbed QBO rather than its initial cause. Nevertheless, more research into the specific role of the SMC of the QBO in modulating the observed QBO modification to SAM is clearly needed.~~ However, the reason for these differences is again the degree of tropical confinement of the aerosol-induced warming of the lower stratosphere because the modification of the residual circulation does ultimately also depend on the meridional gradient of the aerosol-induced temperature anomaly. This shows that using thermal wind balance as the overriding concept to explain the observed differences in the QBO response is reasonable. An increase of the injection rate from 5 Tg(S) yr$^{-1}$ to 25 Tg(S) yr$^{-1}$ as well as an injection of AM−SO$_4$ instead of SO$_2$ act to strengthen the specific QBO response of all three injection strategies.  This has been shown to be a consequence of the stronger warming of the tropical lower stratosphere relative to the subtropical one for the point and region injections, which results in a stronger westerly anomaly.

*Furthermore, we noticed that we referred to a wrong study in Lines 474-475 of the preprint (Section 5), which we corrected in the revised manuscript:*
 Schirber et al. (2014) exemplarily showed for the  ECHAM6 model (Stevens et al., 2013)  that the simulated GW forcing in a GCM is highly dependent on the chosen GW parameterization.

**Review 1:**

**"Major comment: The authors attributed the equatorial winds to the temperature curvature, ..."**
*We are of course aware of the fact that the thermal wind balance is a diagnostic relationship and, therefore, does basically not allow for conclusions about the causal direction of changes in wind and temperature. However, there is no doubt that one field must change as soon as the other field changes, since thermal wind balance must always be satisfied on reasonable time scales. In our opinion this yields the classic question whether the chicken or the egg was first.*
*The reviewer probably agrees with us that the lower stratospheric heating due to the absorption of incoming near infrared and outgoing terrestrial longwave radiation by the artificial sulfate aerosol layer is the initial and major thermal perturbation. Therefore, it must be the root cause of all subsequent changes in the temperature and wind field. Like the reviewer has suggested, this is clearly highlighted when plotting the heating rates, which we have done: We modified Figure 2 in the revised manuscript (Figure 2 in the preprint) by adding 3 subplots of the zonal mean net aerosol heating rate of the experiments point-so2-25, region-so2-25, and 2point-so2-25. We changed the figure caption accordingly:*

Latitude-height cross section of the zonal mean net aerosol heating rate (left column) and of the anomaly of the zonal mean temperature T (right column) for the ECHAM simulations of point-so2-25 (a)(top row), region-so2-25 (b)(middle row), and 2point-so2-25 (c)(bottom row). Stippling indicates areas where the temperature anomalies are not significant at the 95 % level in a student's t-test. Black contour lines indicate the anomaly of the zonal mean sulfate mass mixing ratio $m_{SO4}$ in intervals of 20 $\mu g(S)\ kg^{-1}$.

*Due to the modification of Figure 2 of the revised manuscript (Figure 2 of the preprint), we changed two text passages in Section 3.1.3 of the revised manuscript (Section 3.1 of the preprint):*
The reason for the weaker westerlies is the meridionally more uniform warming of the lower tropical stratosphere aerosol heating in region-so2-25 compared to point-so2-25 (Fig. 2 b) (Fig. 2 c), which results in a meridionally more uniform temperature anomaly within the lower tropical stratosphere (Fig. 2 d).

Due to the extratropical injection locations, the highest sulfate concentrations are located at approximately 20°N and 20°S (Fig. 2 e). and so is the associated heating (Fig. 2 e). Therefore, the stratospheric temperature anomaly is meridionally nearly uniform between approximately 20°N and 20°S (Fig. 2 c) (Fig. 2 f).

*Additionally we also changed some references to Figure 2 (Figure 2) accordingly throughout the manuscript.*

*In order to emphasize more clearly that the aerosol-induced heating of the lower stratosphere and the associated positive temperature anomaly are the root causes of all subsequent changes in stratospheric dynamics, we added Section 3.1.1 "Aerosol-induced heating of the lower stratosphere" to the revised manuscript:*

The artificial sulfate aerosols heat the lower stratosphere by the absorption of OTLR and NIRR, whereby the location and magnitude of this heating strongly correlate with those of the sulfate mass mixing ratio $m_{SO4}$ (Fig. 2 a,c,e). For an equatorial point injection (Fig. 2 a), the sulfate aerosols are strongly concentrated within the tropics, which leads to a strong heating of the lower tropical stratosphere peaking at the equator. In contrast, the sulfate aerosols are distributed meridionally more uniform for the region injection and even more so for the 2point injection (Fig. 2 c,e), which also results in a meridionally more uniform heating for the region and the 2point injection than for the point injection.

This aerosol-induced heating results in a significant positive temperature anomaly centered at the equator (Fig. 2 b,d,f). Following the meridional structure of the net aerosol heating rates, the lower stratospheric temperature anomaly has a clear equatorial peak for the point injection and its poleward gradients are sharp (Fig. 2 b). For the region injection, the lower stratospheric temperature anomaly still peaks at the equator, but with a smaller absolute magnitude; leading to a smaller poleward gradient (Fig. 2 d). For the 2point injection, the temperature anomaly is meridionally nearly uniform between 15°N and 15°S (Fig. 2 f).

The warming of the lower stratosphere is the primary perturbation induced by the sulfate aerosols, as indicated by the good agreement of the sulfate mass mixing ratio, the net aerosol heating rates, and the temperature anomalies. All changes in dynamics – including the QBO – are obviously induced by this initial warming in a second step. Opposite to the lower stratospheric warming, statistically significant negative temperature anomalies are located in the middle and upper tropical stratosphere for all three injection strategies (Fig. 2 b,d,f). However, Figure 2 clearly shows that this cooling is not induced by the radiative effects of the aerosols as it is located well above the aerosol layer and does not match with the net aerosol heating rates. Consequently, these negative temperature anomalies have been induced dynamically due to an increased tropical upwelling (see Aquila et al. (2014)). Therefore, they cannot be seen as a root cause of any change in the QBO.

*Accepting that the strong positive temperature anomaly in the lower stratosphere is the initial perturbation, there is no doubt that the stratospheric wind field has to adopt to it in a way that thermal wind balance still holds since it is in fact also valid in the tropics. Therefore, we think that it is reasonable to assume causality at this point and to interpret the westerly anomaly centered at 20hPa for ECHAM-point-so2-25 in Figure 13 a,b (Figure 12 a,b of the preprint) and centered at 40hPa for ECHAM-region-so2-25 in Figure 13 c,d (Figure 12 c,d of the preprint) in first order as the response of the lower stratospheric wind field to the initial aerosol-induced temperature anomaly via thermal wind. This argumentation is clearly presented in Lines 262-269 of the revised manuscript (Lines 170-178 of the preprint).*

*We agree to the reviewer that it is hardly possible to assume causality between temperature and wind changes above 20hPa in either way. As argued by the reviewer and as visible in the heating rates in Figure 2 of the revised manuscript (Figure 2 of the preprint), changes in the upper stratospheric temperature field are not induced by the aerosol heating and, therefore, must have been induced dynamically. However, we never made the statement for the ECHAM model that the temperature changes above 20hPa are causing a QBO modification via thermal wind. Nevertheless, in order to prevent further misunderstanding, we added a paragraph to Sec. 3.1.3 of the revised manuscript, in*

*which we explicitly state that upper stratospheric temperature changes are induced dynamically and, therefore, one can't assume them to be a root cause of the QBO disruption:*

As discussed in Section 3.1.1, the negative temperature anomalies above approximately 20 hPa have been induced dynamically due to an increased tropical upwelling and negative temperature advection (see Aquilla et al., 2014). While the corresponding anomalies of $T_y$ and $T_{yy}$ above 20hPa are of course still in thermal wind balance with the upper stratospheric wind field (Fig. 5), it is important to mention that this agreement does not imply causality in the way that these upper stratospheric temperature anomalies have caused the lower stratospheric westerly anomaly and QBO modification.

**Minor comments:**
**"Line 96-97: The radiative heating of sulfate...":**
*Of course, also the aerosol's longwave radiative effect is recognized by the radiation scheme of ECHAM, but we missed to state it explicitly in this sentence. We have modified the sentence accordingly in the manuscript:*
The HAM aerosol model couples back to the dynamics by the aerosol optical properties in the shortwave, longwave and near infrared range, which enter the radiative transfer scheme in MAECHAM5 and thus influence the temperature.

**"Line 118: These AMIP simulations...":**
*In this sentence we used the word "simulate" in a wrong way, which caused a misunderstanding. The reviewer is of course right and our AMIP-style simulations can't simulate the surface cooling by the sulfate layer as the temperature of the sea surface is prescribed. What we meant to say is that the usage of the monthly climatological SST values of the period 1988 to 2007 should mimic or rather approximate the expected sea surface cooling due to the sulfate aerosol layer since this cooling is not calculated interactively. Additionally, the fixed SST reduces uncertainties between both models, which would result out of the direct simulation of the surface cooling due to e.g. different radiation schemes. We have changed the sentence accordingly:*
This combination of GHG and SST  forcing allows to  approximately mimic the surface cooling that would be produced by the sulfate layer, while having a consistent surface field for all models and thus removing the source of uncertainty derived from differences in the simulated cooling amongst models.
*This should prevent further misunderstanding. Additionally, the reviewer is right, and these simulations are not suitable for estimating the surface cooling but only the radiative forcing.*

**"Line 207-210: T/L^2 basically provides...":**
*We agree to the reviewer's comment and modified the sentence in the revised manuscript. The point we want to make here is that one can't use equation (3) to estimate the strength of change of $u_z$ using the same scaling L for different injection strategies. As demonstrated by our comparison between point-so2-5 and 2point-so2-25 in Lines 291-295 of the revised manuscript (Lines 203-207 of the preprint), L obviously depends on the injection strategy. This may seem self-explanatory for some readers, but we thought it's worth mentioning explicitly in order to raise attention to not use equation (3) carelessly.*

*When using one and the same injection strategy, equation (3) is a good approximation for estimating the thermal wind response to different injection rates, as L is roughly constant for one and the same injection strategy.*

This comparison shows that within $u_z \sim R(H\beta)^{-1}L^{-2}T$ (3), which is often used as an approximation of thermal wind balance for QBO variations centered at the equator (Baldwin et al., 2001),  the scaling factor L depends on the injection strategy. Therefore, Equation 3 cannot be used when comparing the QBO response to different injection strategies.

**"L 251: Fig. 2 does not show...":**
*We have set an erroneous Figure reference in Latex. We intended to refer to Figure 6 of the revised manuscript (Figure 4 of the preprint) and changed the reference accordingly.*

**"L 397-401: The authors attributed the ozone...":**
*A more in-depth analyses of the effects of sulfate aerosols at higher altitudes has been performed in Tilmes et al. (2018) and Richter et al. (2017). In particular, in the former, for a case with injection altitudes much similar to ours (case LOW) we can observe that there's no real visible increase of $SO_4$ mixing ratio (due to wide contours, see Fig. 5 in that paper), but given the negligible amount of $SO_4$ in normal conditions, even a slight increase may bring $NO_x$ loss (see Fig. 15 in the referenced paper). Also by looking at Fig. 13 in the same paper (third row) it is clear that there are two overlapping effects of chemical loss and advection resulting in the $NO_x$ changes (and the sequent changes in ozone) that we illustrate in this work.*

*Additionally, the absolute ozone concentration peaks in the middle stratosphere between 10hPa and 5hPa and is much higher in this altitude compared to the one in the lower stratosphere around 30hPa. Therefore, even small changes in $NO_x$ in the middle to upper stratosphere due to sulfate aerosols have a large impact on the ozone concentration.*

*Nevertheless, we understand the reviewer's point as it indeed looks like the sulfate layer does not reach above 15 hPa based on the contour interval chosen in Figure 14 of the preprint. Therefore, we have added additional contours in Figure 15 of the revised manuscript (Figure 14 of the preprint) and changed the figure caption accordingly:*

Latitude-height cross section of the ozone concentration anomaly for the simulations of an AM−SO$_4$ injection in CESM. Stippling indicates regions where anomalies are not significant at the 95 % level in a student's t-test. Contour lines indicate the zonal mean sulfate mass mixing ratio $m_{SO4}$ in µg kg$^{-1}$. The contour interval is logarithmic starting at 25  µg kg$^{-1}$. The sulfate mass mixing ratio can be reasonably used as a proxy for the heating rate due to absorption of OTLR and NIRR.

*With regard to the second part of this comment (**"It is also interesting to compare..."**):*
*The vertical structure of the ozone changes is indeed very similar between CESM-region-so4 and CESM-region-so4-25. At this point we would again refer to the in-depth analysis by Tilmes et al. (2018): As visible in the lowermost panels in Fig. 15 in the referenced paper, the loss rates in the high and low injection case are – relatively spoken – rather similar, indicating that the vertical structure of the ozone changes is rather independent of the vertical extent of the aerosol layer.*
*Additionally, also the temperature dependence of the injection rates may play a role here because*

*reaction R1 is more efficient at colder temperatures, and therefore in the lower stratosphere. Therefore, even though there are higher aerosol concentrations at higher altitudes in CESM-region-so4-25 than in CESM-region-so4-5, ozone depletion stops above roughly 20hPa since temperatures are too high for R1 working efficiently. However, the effect of the temperature dependence is minor compared to other effects because that high up the cooling effect from $CO_2$ mostly balances out the warming.*

*As our paper focuses on the QBO response to artificial sulfur injections, we decided to not add these details into the manuscript as it would go beyond the scope of the paper, but to only add a reference to Tilmes et al. (2018), which examined the ozone changes due to sulfur injections in more detail, so the interested reader can find more information:*
For more details on stratospheric ozone depletion, we refer to Tilmes et al. (2018).

**References:**

Richter, J. H., Tilmes, S., Mills, M. J., Tribbia, J. J., Kravitz, B., MacMartin, D. G., Vitt, F., and Lamarque, J.-F.: Stratospheric Dynamical Response and Ozone Feedbacks in the Presence of SO2 Injections, Journal of Geophysical Research: Atmospheres, 122, 12,557–12,573, https://doi.org/10.1002/2017JD026912, 2017.

Tilmes, S., Richter, J. H., Mills, M. J., Kravitz, B., MacMartin, D. G., Garcia, R. R., Kinnison, D. E., Lamarque, J.-F., Tribbia, J., and Vitt, F.: Effects of Different Stratospheric SO2 Injection Altitudes on Stratospheric Chemistry and Dynamics, Journal of Geophysical Research: Atmospheres, 123, 4654–4673, https://doi.org/10.1002/2017JD028146, 2018.

**Review 2:**

*First of all, we would like to thank the reviewer for his/her detailed and valuable feedback.*
*The reviewer wrote the following sentence in his/her summary:* **"They finally demonstrate that ozone plays a role in why the injected aerosol type matters; (...)".**
*This must have been a misunderstanding, since the fact that ozone is likely to play a key role in determining the QBO response to SAM is independent of our tested injection species and occurred for both, $SO_2$ and AM-$SO_4$.*
*Figure S5 in the supplementary material corresponds to Figure 12 of the revised manuscript (Figure 11 of the preprint) and shows the QBO response to the corresponding $SO_2$ injections. The qualitative characteristics of both plots agree, since the injection species only matters for the QBO response quantitatively as discussed in Section 3.3 of the revised manuscript (Sec. 3.4 of the preprint). Therefore, we compared ECHAM and CESM only for the injection of AM-$SO_4$, as stated in Lines 357-360 of the revised manuscript (Lines 295-298 of the preprint). Nevertheless, our conclusion that an interactive ozone chemistry is likely very important for the simulated QBO response to artificial sulfur injections does of course also hold for an injection of $SO_2$. To highlight this and to prevent further misunderstanding, we added the following sentence to Section 4.3 of the revised manuscript:*
These considerations also hold true for the region injections of $SO_2$, which resulted in qualitatively similar ozone changes and modifications of the QBO (see supplementary material).

*Additionally, the reviewer has mentioned similar doubts regarding the causality issue of thermal wind balance like reviewer 1:* **"Overall, they claim that arguments related to thermal wind balance are sufficient to explain the changes and sensitivities that are found, although I do have doubts about how strong their causation arguments are using this inherently diagnostic relationship (...)".**
*Therefore, regarding this comment, we refer to our answer to the major comment of reviewer 1, which addresses this point:*

**"***We are of course aware of the fact that the thermal wind balance is a diagnostic relationship and, therefore, does basically not allow for conclusions about the causal direction of changes in wind and temperature. However, there is no doubt that one field must change as soon as the other field changes, since thermal wind balance must always be satisfied on reasonable time scales. In our opinion this yields the classic question whether the chicken or the egg was first.*
*The reviewer probably agrees with us that the lower stratospheric heating due to the absorption of incoming near infrared and outgoing terrestrial longwave radiation by the artificial sulfate aerosol layer is the initial and major thermal perturbation. Therefore, it must be the root cause of all subsequent changes in the temperature and wind field. Like the reviewer has suggested, this is clearly highlighted when plotting the heating rates, which we have done: We modified Figure 2 in the revised manuscript (Figure 2 in the preprint) by adding 3 subplots of the zonal mean net aerosol heating rate of the experiments point-so2-25, region-so2-25, and 2point-so2-25. We changed the figure caption accordingly:*
Latitude-height cross section of the zonal mean net aerosol heating rate (left column) and of the anomaly of the zonal mean temperature T (right column) for the ECHAM simulations of point-so2-25

(top row), region-so2-25 (middle row), and 2point-so2-25 (bottom row). Stippling indicates areas where the temperature anomalies are not significant at the 95 % level in a student's t-test. Black contour lines indicate the anomaly of the zonal mean sulfate mass mixing ratio m$_{SO4}$ in intervals of 20 µg(S) kg$^{-1}$.

*Due to the modification of Figure 2 of the revised manuscript (Figure 2 of the preprint), we changed two text passages in Section 3.1.3 of the revised manuscript (Section 3.1 of the preprint):*
The reason for the weaker westerlies is the meridionally more uniform  aerosol heating in region-so2-25 compared to point-so2-25  (Fig. 2 c), which results in a meridionally more uniform temperature anomaly within the lower tropical stratosphere (Fig. 2 d).

Due to the extratropical injection locations, the highest sulfate concentrations are located at approximately 20°Nand 20°S  and so is the associated heating (Fig. 2 e). Therefore, the stratospheric temperature anomaly is meridionally nearly uniform between approximately 20°Nand 20°S  (Fig. 2 f).

*Additionally, we also changed some references to Figure 2 (Figure 2) accordingly throughout the manuscript.*

*In order to emphasize more clearly that the aerosol-induced heating of the lower stratosphere and the associated positive temperature anomaly are the root causes of all subsequent changes in stratospheric dynamics, we added Section 3.1.1 "Aerosol-induced heating of the lower stratosphere" to the revised manuscript:*
The artificial sulfate aerosols heat the lower stratosphere by the absorption of OTLR and NIRR, whereby the location and magnitude of this heating strongly correlate with those of the sulfate mass mixing ratio m$_{SO4}$ (Fig. 2 a,c,e). For an equatorial point injection (Fig. 2 a), the sulfate aerosols are strongly concentrated within the tropics, which leads to a strong heating of the lower tropical stratosphere peaking at the equator. In contrast, the sulfate aerosols are distributed meridionally more uniform for the region injection and even more so for the 2point injection (Fig. 2 c,e), which also results in a meridionally more uniform heating for the region and the 2point injection than for the point injection.
This aerosol-induced heating results in a significant positive temperature anomaly centered at the equator (Fig. 2 b,d,f). Following the meridional structure of the net aerosol heating rates, the lower stratospheric temperature anomaly has a clear equatorial peak for the point injection and its poleward gradients are sharp (Fig. 2 b). For the region injection, the lower stratospheric temperature anomaly still peaks at the equator, but with a smaller absolute magnitude; leading to a smaller poleward gradient (Fig. 2 d). For the 2point injection, the temperature anomaly is meridionally nearly uniform between 15°N and 15°S (Fig. 2 f).
The warming of the lower stratosphere is the primary perturbation induced by the sulfate aerosols, as indicated by the good agreement of the sulfate mass mixing ratio, the net aerosol heating rates, and the temperature anomalies. All changes in dynamics – including the QBO – are obviously induced by this

initial warming in a second step. Opposite to the lower stratospheric warming, statistically significant negative temperature anomalies are located in the middle and upper tropical stratosphere for all three injection strategies (Fig. 2 b,d,f). However, Figure 2 clearly shows that this cooling is not induced by the radiative effects of the aerosols as it is located well above the aerosol layer and does not match with the net aerosol heating rates. Consequently, these negative temperature anomalies have been induced dynamically due to an increased tropical upwelling (see Aquila et al. (2014)). Therefore, they cannot be seen as a root cause of any change in the QBO.

*Accepting that the strong positive temperature anomaly in the lower stratosphere is the initial perturbation, there is no doubt that the stratospheric wind field has to adopt to it in a way that thermal wind balance still holds since it is in fact also valid in the tropics. Therefore, we think that it is reasonable to assume causality at this point and to interpret the westerly anomaly centered at 20hPa for ECHAM-point-so2-25 in Figure 13 a,b (Figure 12 a,b of the preprint) and centered at 40hPa for ECHAM-region-so2-25 in Figure 13 c,d (Figure 12 c,d of the preprint) in first order as the response of the lower stratospheric wind field to the initial aerosol-induced temperature anomaly via thermal wind. This argumentation is clearly presented in Lines 262-269 of the revised manuscript (Lines 170-178 of the preprint).*

*We agree to the reviewer that it is hardly possible to assume causality between temperature and wind changes above 20hPa in either way. As argued by the reviewer and as visible in the heating rates in Figure 2 of the revised manuscript (Figure 2 of the preprint), changes in the upper stratospheric temperature field are not induced by the aerosol heating and, therefore, must have been induced dynamically. However, we never made the statement for the ECHAM model that the temperature changes above 20hPa are causing a QBO modification via thermal wind. Nevertheless, in order to prevent further misunderstanding, we added a paragraph to Sec. 3.1.3 of the revised manuscript, in which we explicitly state that upper stratospheric temperature changes are induced dynamically and, therefore, one can't assume them to be a root cause of the QBO disruption:*

As discussed in Section 3.1.1, the negative temperature anomalies above approximately 20 hPa have been induced dynamically due to an increased tropical upwelling and negative temperature advection (see Aquilla et al., 2014). While the corresponding anomalies of $T_y$ and $T_{yy}$ above 20hPa are of course still in thermal wind balance with the upper stratospheric wind field (Fig. 5), it is important to mention that this agreement does not imply causality in the way that these upper stratospheric temperature anomalies have caused the lower stratospheric westerly anomaly and QBO modification."

*Below you'll find our answers to the detailed minor and technical comments put forth by the reviewer, which we really appreciated and tried to consider carefully.*

**"Lines 31-32: Can you state explicitly here...":** *Labitzke (1994) did not provide reasoning. However, the reviewer is right, and it is very likely that an increased tropical upwelling due to the aerosol-induced stratospheric temperature anomaly (Giorgetta et al., 2011). We added the following modified sentence to the revised manuscript:*

After the major eruption of Mt. Pinatubo in June 1991, the lower stratosphere warmed by about 3K, which led to a prolonged QBO westerly phase in the lower stratosphere (Labitzke, 1994), very likely due to an increased tropical upwelling induced by the aerosol warming (Giorgetta et al., 2011).

**"Line 44; 'further' --> 'also'":** *Done.*

**"Line 54; Not sure what this sentence means. ...":** *The reviewer is right and we mean that the QBO response in both models was qualitatively similar but quantitatively different. We changed the sentence in the revised manuscript:*
Both models showed  a qualitatively similar QBO response on SAM, but quantitatively much stronger in WACCM-110L.

**"Line 75; delete 'one'":** *Done.*

**"Line 85; which parameterisation? ...":** *MAECHAM5 uses a gravity wave parameterization following Hines (1997a, b) and its implementation in MAECHAM5 is described by Manzini (2006). Since our study doesn't focus on the role of gravity waves in the QBO modification, we decided to only refer to the relevant publications instead of giving a more detailed description of the parameterization in the manuscript itself. Therefore we added the following sentence to the revised manuscript:*
Additionally, it accounts for the momentum flux deposition of unresolved gravity waves (GW) originating from the troposphere via a parameterization following Hines (1997a, b); its implementation into MAECHAM5 is described by Manzini et al. (2006).

**"Line 124; Why is CESM not ...":** *We have addressed this question further on in the preprint in Lines 298-299 (Lines 360-361 of the revised manuscript):* "A comparison of the QBO response to the point injection strategy is not possible, since the point injection is no part of the accumH2SO4experiment protocol and was, therefore, not performed by CESM."
*For ECHAM the point injections have been simulated in addition to the 2point and region injections which were prescribed in accumH2SO4, but we did not have enough computational resources to do the same for CESM. Nevertheless, we think it is very important to include the CESM simulations in this paper because of the significant differences between the two models.*
*However, we agree to the reviewer and think it is worthwhile to mention the reason for the lack of the missing CESM point simulations already in Section 2.2. Therefore, we have added the following sentence to the manuscript:*
The point injection strategy is not part of the accumH2SO4 experiment protocol and was, therefore, not performed by CESM.

**"Table 1; It needs to be stated...":** *For all three injection strategies the total injected aerosol load is identical. To stay with the example of the reviewer, the 2point-25 experiments inject 12.5Tg(S)/yr at each injection point, which results in the total injected amount of 25Tg(S)/yr. This allows for the direct comparison of the tested injection strategies. To prevent further misunderstanding, we added the following sentence in Section 2.3:*

The injection rate is the total amount of sulfur that is injected globally per year; for example, in the 2point injections with an injection rate of 25Tg(S)yr$^{-1}$, each of both injection points has an injection rate of 12.5Tg(S)yr$^{-1}$.

*Furthermore, we added the following sentence to the caption for Table 1:*
The injection rate is the total amount of sulfur that is injected globally per year.

*Additionally, the reviewer is right and to be exactly precise one should have taken the smaller grid box size further poleward into account. Nevertheless, for MAECHAM5 the size of the grid box centered at 29.3°N or 29.3°S (2point injection) is only approx. 13% smaller than the one of the grid box centered at 1.4°N (point injection). In our opinion this small difference is neglectable, especially given the fact that in reality a point-like injection would be neither a perfect point-like injection nor an injection into an area defined by a grid box but something in between.*

**"Lines 131-133; I think a bit of background..."**: *Our simulations contributed to the GeoMIP6 experiment accumH2SO4, which prescribed the injection rates of 5Tg(S)/yr and 25Tg(S)/yr. 5Tg(S)/yr is a rough approximation of the amount of sulfur injected as SO$_2$, which would be necessary to counteract a greenhouse gas (GHG) forcing of 1.5 W/m$^2$ as prescribed in the RCP4.5 scenario (see e.g. Niemeier et al., 2013) and is therefore commonly used. However, to reduce a strong GHG forcing like the RCP8.5 scenario, much higher injection rates are necessary. An injection rate of 25Tg(S)/yr is commonly used to resemble these high injection scenarios.*
*We amended the sentence in Lines 134-135 of the revised manuscript (Line 131 of the preprint):*
All injection scenarios have been simulated with two different injection rates for both models: 5 and 25Tg(S) yr$^{-1}$, as given in the accumH2SO4 protocol.

*However, since we don't know the exact reason why these injection rates have been chosen by the responsibles of the accumH2SO4 project, we decided not to add further details to the manuscript.*

**"Section 3.1; The word 'disruption' in the..."**: *We agree to the reviewer that the word 'disruption' is a misnomer. Of course thermal wind balance still holds and is, therefore, not disrupted but rather modified. We changed the title of Section 3.1.3 of the revised manuscript (Section 3.1 of the preprint) accordingly:*
 Modification of thermal wind balance

**"Line 172; why is the experiment chosen..."**: *We chose the experiments of a SO$_2$ injection of 25Tg(S)/yr because of two reasons. The reason for choosing an injection of SO$_2$ instead of an injection of AM-SO$_4$ at this point was that the great majority of previous studies on artificial sulfur injections has focused on an injection of SO$_2$, especially those investigating the QBO response to artificial sulfur injections including the pioneering study by Aquilla et al. (2014). Thereby, we wanted to assure a certain degree of comparability to those earlier studies at this point.*
*The reason for choosing the experiments with an injection rate of 25Tg(S)/yr instead of those with an injection rate of 5Tg(S)/yr was the stronger signal-to-noise ratio for a higher injection rate.*

*In order to highlight this in more detail, we added the following sentences to Section 3.1 of the revised manuscript:*

In this Section, we will investigate the reasons for the different QBO response to the three tested injection strategies exemplarily based on an injection of $SO_2$ with an injection rate of 25Tg(S) $yr^{-1}$(experiments point-so2-25, region-so2-25,2point-so2-25). This injection scenario follows the experimental setup of Aquila et al. (2014) with regard to the injection type and has a high signal-to-noise ratio due to the high injection rate. The impact of a lower injection rate and another injection species (i.e. $AM-SO_4$ instead of $SO_2$) will be discussed in Sections 3.2 and 3.3.

*As shown in Figure 6 of the revised manuscript (Figure 4 of the preprint), one can't assume perfect linearity for the lower stratospheric temperature anomalies for an injection of SO₂. Increasing the injection rate by a factor of 5 (from 5Tg(S)/yr to 25Tg(S)/yr) only yields a 4 times stronger tropical lower stratospheric temperature anomaly. The same holds true for an injection of AM-SO₄ (Fig. S1 in the supplementary material). Nevertheless, as this is quite close to linearity, we think that our results in Section 3.1 of the revised manuscript, which are based on the 25Tg(S)/yr experiments, can be downscaled reasonably well to the experiments with an injection rate of 5Tg(S)/yr.*

*Additionally, we changed* 'exemplary' *to* 'exemplarily' *throughout the manuscript.*

**"Line 174; what does 'usually positive poleward' mean here? ...":** *Under "normal" conditions (i.e. in contr-000), the tropical tropopause region and the lower tropical stratosphere are colder than the subtropical or extratropical lower stratosphere (40hPa to 160hPa) as shown in Figure 4 of the revised manuscript. Therefore, starting at the equator, lower stratospheric temperatures on average increase in the poleward direction (no matter if southward or northward) until approximately 50°S and 50°N, respectively. Consequently, the poleward temperature gradient is usually positive in the lower stratosphere. This was meant by the term "usually positive poleward $T_y$". To prevent future misunderstanding, we have added a plot of the climatological mean conditions (Fig 4 of the revised manuscript) and highlighted the climatological mean conditions in Section 3.1.3 of the revised manuscript:*

As discussed in Section 3.1.1, an equatorial point injection results in a significant positive temperature anomaly centered at the equator (Fig. 2 b). In the climatological mean – without artificial sulfur injections and represented by contr-000 – the lower tropical stratosphere is much colder than the lower midlatitudinal stratosphere (Fig. 4), leading to a poleward $T_y$ that is positive. The aerosol-induced warming abates the  poleward $T_y$ (Fig. 5 a) within the lower tropical stratosphere between 40hPa and 80hPa , which is accompanied by a significant negative anomaly of $T_{yy}$ centered at the equator  (Fig. 5 b). Following Equation 2, a negative anomaly of $T_{yy}$ results in stronger westerly shear. Consequently, a strong westerly anomaly of the zonal mean zonal wind u is located on top of the injection layer in order to maintain thermal wind balance  (Fig. 5 b). This results in the observed constant westerly QBO phase (Fig. 1 b).

*The anomaly of the temperature gradient is calculated in the same way as all other anomalies and as stated in Lines 141-142 of the revised manuscript (Lines 136-137 of the preprint). So, in Figure 5 of the*

*revised manuscript (Figure 3 of the preprint), the temperature gradient anomaly is calculated as the difference between the temporally averaged temperature gradient of contr-000 and the injection experiments. The same holds true for the calculation of the temperature curvature anomaly.*

**"Line 183-184; Does figure 2b really show...":** *In our opinion it does. When staying at one pressure level in the aerosol layer between 40hPa and 80hPa and drawing a horizontal line at that level into Figure 2b and 2d of the revised manuscript, one clearly sees that the meridional gradient of the temperature anomaly at that level is weaker in Figure 2d than in Figure 2b based on the given contour levels. We interpret this as a meridional more uniform warming of region-so2-25 (Fig. 2d) relative to point-so2-25 (Fig.2b), while spoken in absolute terms the warming is of course also not meridionally uniform in Figure 2d. Therefore, we decided not to change anything in the manuscript.*

**"Line 185; still in the lower stratosphere...":** *We agree to the reviewer that at this point we should mention that this argument only holds for the lower stratosphere in order to prevent misunderstanding. We modified the manuscript:*

Therefore, the strongest modifications of $T_y$ in the lower stratosphere are located poleward of approximately 20°Nand 20°S, while its modification close to the equator is relatively small (Fig. 5 c).

**"Lines 196-197; I do not see this slight intensification...":** *We agree to the reviewer that this intensification is hardly statistically significant and, therefore, we can't make such strong arguments as in Lines 193-201 of the preprint. Therefore, we removed this text passage and modified the paragraph in the revised manuscript:*

For 2point-so2-25, the QBO was not found to be modified significantly and it basically preserved its natural periodicity (Fig. 1 k). Due to the extratropical injection locations, the highest sulfate concentrations are located at approximately 20°N and 20°S and so is the associated heating (Fig. 2 e) (Fig. 2 c). Therefore, the lower-stratospheric temperature anomaly is meridionally nearly uniform between approximately 20°N and 20°S(Fig. 2 c) 15°N and 15°S (Fig. 2 f). Consequently, $T_y$ as well as $T_{yy}$ are basically not only very weakly modified within the lower tropical stratosphere between 40hPa and 80hPa and between approximately 15°S and 15°N (Fig. 5 e,f) (Fig. 3 e,f) and these anomalies are hardly statistically significant. Following thermal wind balance, this explains why the QBO remains in principle in its natural shape. implies that zonal wind anomalies in the lower stratosphere are small as well (Fig. 5 e,f), which is in accordance with the QBO in principle remaining in its natural shape. When looking in more detail, the lower stratospheric temperature anomaly even has a slightly convex shape between approximately 15°N and 15°S for the 2point injections (Fig. 2 c), which is in contrast to the point and region injection strategies. Therefore, the usually positive poleward $T_y$ in the lower stratosphere (40 hPa to 80 hPa) slightly intensifies between approximately 15°S and 15°N (Fig. 3 e). This leads to a slight positive anomaly of $T_{yy}$ centered at the equator and approximately 50 hPa (Fig. 3 f) and is accompanied by an anomaly of easterly shear following Equation 2. Accordingly, u has a slight easterly anomaly above the injection layer in 2point-so2-25 (Fig. 3 bottom). The consequence is a reduction of the asymmetry between the westerly and easterly QBO phases resulting (Fig. 1 bottom). However, this slight speed up of the QBO phase is not significant based on our short simulation period.

**"Lines 170-201; I think you need to be careful...":** *This concern of reviewer 2 very much agrees to the major concern of reviewer 1, which we already included into this reply to reviewer 2 earlier. Therefore, at this point we refer to the beginning of this reply to reviewer 2, in which we state why at least in the lower stratosphere causation can be reasonably assumed.*

**"Lines 203-206: This is true when comparing...":** *The reviewer is right with his/her statement and our formulation in the preprint was not clear enough. We modified the corresponding text passage in the manuscript:*

Therefore, the absolute strength of the aerosol-induced lower stratospheric temperature anomaly does not permit a statement about the strength of the QBO modification when comparing different injection strategies.

*As we wrote earlier in this reply, there is no precise linearity between the injection strength and the temperature response; and the same holds true for the wind anomaly.*

*However, the overall injection magnitude is identical for all three injection strategies, and, therefore, a direct comparison of all three injection strategies is reasonable.*

**"Lines 207-210; I do not follow this. ...":** *Reviewer 1 had some concerns regarding our statement in Lines 207-210 of the preprint as well, which was caused by an imprecise formulation. We have modified the sentences in the revised manuscript, which should also help reviewer 2 to understand this more easily:*

This comparison shows that within $u_z \sim R(H\beta)^{-1} L^{-2} T$ (3), which is often used as an approximation of thermal wind balance for QBO variations centered at the equator (Baldwin et al., 2001),  the scaling factor L depends on the injection strategy. Therefore, Equation 3 cannot be used when comparing the QBO response to different injection strategies.

**"Lines 223-224; You only state this observation...":** *The reason for the upper stratospheric w\* anomalies being much stronger than the lower stratospheric anomalies of w\* was given a few lines below in Lines 234-244 of the preprint (Lines 224-227 of the revised manuscript): These strong upper stratospheric w\* anomalies are a consequence of the fact that also the SMC of the QBO is locked in its "westerly" phase due to the permanent lower stratospheric westerlies (see Plumb & Bell, 1982 or Punge et al., 2009 for details on the SMC). Following e.g. Figure 1a of Punge et al. (2009), the strongest upwelling by the SMC is located where the westerlies change to easterlies, which is also exactly the case in our simulations (compare Figure 1e and 3a of the revised manuscript, Fig. 1e and 3a of the preprint). This explains why the strongest w\* anomalies are located well above the aerosol layer.*

*In order to increase comprehensibility of our explanation of the observed w\* increase, we have restructured the corresponding text passage:*

The reason for the increase of  w\* is the aerosol-induced stratospheric temperature anomaly, which alters  the characteristics of the zonal jets in the extratropical stratosphere. Thereby, the conditions for the vertical propagation of planetary waves in this region change. As a

consequence, the extratropical wave-driving of the residual mean circulation increases, which ultimately speeds up the whole BDC. This mechanism has been investigated by Tilmes et al. (2018) for SAM simulations and was also recognized in simulations of a tropical volcanic eruption by Bittner et al. (2016).

 In the upper stratosphere (i.e. between 20hPa and 3hPa in point-so2-25 and between 25hPa and 8hPa in region-so2-25), this increase of w* is superimposed by changes of the secondary meridional circulation (SMC) of the modified QBO itself. During a permanent QBO westerly phase, the SMC would also be permanently locked in its corresponding "westerly" phase, which acts to increase w* within the tropical stratosphere (Plumb and Bell, 1982). Our experiments indicate that a large fraction of the increase of w* in the upper tropical stratosphere  (Fig. 3 a,b) can be attributed to this "indirect" acceleration via the SMC, especially in the upper stratosphere. For example, in the experiment point-so2-5, the tropical w* increased by up to 100% in the upper stratosphere (not shown). In contrast, in ECHAM simulations with permanent lower stratospheric easterlies instead of a QBO, Niemeier et al. (2011) obtained an increase of the tropical w* of only 5 to 10% for an equatorial point injection of $SO_2$ with an injection rate of $4Tg(S) yr^{-1}$. Therefore, one has to be cautious when interpreting the strong positive anomalies of $-w^* u_z$ observed in the upper stratosphere in point-so2-25 and region-so2-25 as the primary cause for the disruption of the QBO since they are – at least partly – rather its consequence.

**"Line 230; The static stability changes...":** *We agree to the reviewer in that it is due the changes in winds at higher latitudes that changes in planetary wave driving occur. This was exactly what we intended to say in Lines 229-234 of the preprint (Lines 219-223 of the revised manuscript) and what was espoused by Bittner et al. (2016) and Tilmes et al. (2018). If we have understood the paper correctly, then the mechanism by how the tropical stratospheric temperature anomaly facilitates extratropical wave propagation does approximately follow Chen and Robinson (1992). Nevertheless, we agree that static stability is predominantly modified in the tropics and its impact on the increase of the tropical upwelling is only small. Therefore, in the revised manuscript we don't mention changes of static stability as a reason for a modified extratropical wave driving any longer (see our answer to the previous minor issue brought up by the reviewer).*

**"Lines 237-238; How can you attribute...":** *In Lines 227-234 of the revised manuscript (Lines 239-242 of the preprint) we argue why we attribute this change in w\* mostly to changes in the SMC instead of to just a change in the BDC. Niemeier et al. (2011) observed an increase of the tropical w\* of about 5-10% in an ECHAM simulation of point-so2-4 with a prescribed QBO, and consequently, prescribed SMC. So this increase of w\* must be due to the BDC. In our ECHAM simulations of point-so2-5 with an interactive QBO and SMC, w\* increased by about 10-40% in the lower stratosphere and about 100% in the upper stratosphere. As the injection rates of both experiments are close to each other, the difference in w\* increase between both experiments is very likely due to the SMC.*

**"Line 247; Is there a threshold...":** *We agree to the reviewer that it is of large interest to know*

*whether there is an unambiguous threshold and in case there is one, how large it is. With our limited set of simulations, we are not able to determine it accurately for our three injection strategies. In ECHAM simulations with an injection rate of 10Tg(S)/yr (not shown), the QBO was already locked for an injection rate of 10Tg(S)/yr for the point injections, while for the region injections it was still oscillating, but clearly prolonged. Additionally, the QBO response to SAM is also very dependent on the used model as shown by the comparison with CESM in our manuscript or by Niemeier et al. (2020), which would further complicate the determination of an exact threshold. However, further research into this is urgently needed which includes a better understanding of the differences between the models.*

**"Figure 6; The 5S-5N meridional average...":** *We guess that the reviewer may have confused $T_y$ and $T_{yy}$ in Figure 3. We agree that the anomalies of $T_y$ are antisymmetric about the equator and, therefore, would likely cancel each other in a 5°S-5°N mean as seen in the right column of Figure 5 of the revised manuscript (Figure 3 of the preprint). However, $T_{yy}$ is symmetric about the equator and has the same sign within the 5°S-5°N belt, which is why the anomalies on the northern and southern hemisphere don't cancel each other. Since Figure 7 of the revised manuscript (Figure 6 of the preprint) shows the 5°S-5°N mean of $T_{yy}$ and not of $T_y$, we think it's a reasonable measure at this point.*

**"Figure 7; at what height is this line plot?":** *This line plot shows the artificial burden, which is the vertical integral of the sulfur concentration, an atmospheric column quantity (unit $mg(S)/m^2$). Therefore, this plot is not at a specific height. To help understanding for persons not so familiar with geoengineering, we added the following sentence to the manuscript where the term sulfate burden is used for the first time:*

This is explained by the  tropical sulfate burden, which  is the vertical integral of the sulfur concentration and which is clearly lower for a lower injection rate.

*Additionally, we also changed the caption of Figure 8 of the revised manuscript (Figure 7 in the preprint):*

Zonal mean artificial sulfate burden, which is the vertical integral of the sulfate concentration, for the ECHAM simulations of the $SO_2$ injections (dashed lines) and the $AM-SO_4$ injections (solid lines) with an injection rate of $25Tg(S)yr^{-1}$ as a function of latitude.

**"Lines 271-274; Is there a difference between...":** *We have calculated the significances of the difference between these different injection species runs (point-so2-25 vs. point-so4-25, 2point-so2-25 vs. 2point-so4-25, region-so2-25 vs. region-so4-25). For the point and region injection strategy, the zonal wind difference between an injection of $SO_2$ and $AM-SO_4$ is statistically significant at the 95%-level using a student's t-test between 50hPa and 3hPa. For the 2point injection strategy, the zonal wind difference between an injection of $SO_2$ and $AM-SO_4$ is statistically significant at the 95%-level using a student's t-test only between 33hPa and 20hPa.*
*For the temperature curvature $T_{yy}$ only the levels between 40hPa and 80hPa are relevant since the primary aerosol heating, which causes the thermal wind response, is located in this height range. However, the $T_{yy}$ difference between an injection of $SO_2$ and $AM-SO_4$ is statistically significant at the 95%-level based on a student's t-test only for the point injection strategy between 57hPa and 40hPa.*

*For the 2point and region injection strategy, the differences are not significant. Therefore, the manuscript has been changed correspondingly:*

As described in Section 3.3, the accompanied stronger warming of the lower tropical stratosphere relative to the mid-latitude one results in a stronger modification of $T_{yy}$  (Fig. 9 a). However, the difference of the $T_{yy}$ anomaly between both injection species is statistically significant at the 95 % level in a student's t-test only for the point injections. This causes a stronger QBO westerly phase for an injection of AM−SO$_4$ compared to an injection of SO$_2$ as indicated by the anomalies of u  (Fig. 9 b). The difference of the u anomaly between both injection species is statistically significant at the 95 % level in a student's t-test for both injection strategies, the point and the region injections.

**"Figure 10; the final sentence of caption is...":** *We added a dotted line marking a RF of -4W/m$^2$ to Figure 11 of the revised manuscript (Figure 10 of the preprint) and changed the figure caption accordingly:*

Global mean TOA all-sky net RF exerted by artificial sulfate aerosols as a function of injection rate. The  dotted black line marks a RF of -4Wm$^{-2}$.

*Furthermore, we agree to the reviewer that it would be nice to have experiments with an injection rate of either 10Tg(S)/yr or 15Tg(S)/yr, especially for an injection of SO$_2$ since the forcing doesn't depend linearly on injection rate. However, they couldn't have been performed due to limits in computational resources.*

**"Figure 11; It would be useful if you...":** *Determining an exact value of the QBO period based on our short simulation period in a reasonable way would be hardly possible. Therefore, we decided to not include the QBO periodicity in Figure 12 of the revised manuscript (Figure 11 of the preprint). In the manuscript we never gave an exact value of the QBO period apart from the control runs. For our argumentation the exact value of the QBO period is not important, but only the difference of the QBO period and shape between the different experiments and models. We think that these differences between the different experiments are easy to grasp from Figure 12 of the revised manuscript (Figure 11 of the preprint) also without including the QBO period in each panel.*

**"Figure 12; the thick black denoting u=0...":** *We have plotted Figure 13 of the revised manuscript (Figure 12 of the preprint) with thicker u=0m/s contour lines. The corresponding Figure S6 in the Supplementary Material has been also redone.*

**"Lines 308-309; 'in principle' --> 'qualitatively' ...":** *We see the concerns of the reviewer at this point. Therefore, we decided to refer to Figure 12 in the revised manuscript (Figure 11 in the preprint) instead of to Figure 11 (Figure 10) as done in the preprint. Figure 12 of the revised manuscript (Figure 11 of the preprint) shows the qualitative agreement between the QBO of both models more clearly. Additionally, we decided to build up the following argumentation on the 25Tg(S)/yr experiments only. Therefore, we did the following changes in the manuscript:*

Nevertheless, the QBO  responds qualitatively similar to a 2point injection of AM−SO$_4$ in both models, which is clearly shown by Figures 12  c-f. The spatial structure of the u anomalies, indicated by the contour lines in Figure  13, does in general agree for both models and

both tested injection rates (Fig. 13 a-d). For an injection rate of 25Tg(S)/yr, it shows a lower stratospheric easterly anomaly centered at approximately 40hPa and 5°S and an upper stratospheric westerly anomaly, which further extends into the northern hemisphere.

**"Lines 313-314; In fig. 12c,d I do not agree...":** *With this sentence we meant to reference to the lower stratospheric temperature anomalies between 40hPa and 80hPa and 20°S to 20°N. We think that Figure 13 c,d of the revised manuscript (Figure 12 of the preprint) clearly shows that those particular $T_y$ anomalies are stronger and more poleward extended in CESM than in ECHAM. However, we agree to the reviewer in that it looks the other way around for the temperature anomalies further aloft and that our sentence was not precise enough in the preprint. Therefore, we changed the sentence in the revised manuscript:*
For a given injection rate, this strengthening – located approximately between 80hPa and 40hPa and between 20°S and 20°N – is clearly stronger and vertically more extended in CESM (Fig. 12 c,d) (Fig. 13 c,d), which explains the stronger easterly anomaly observed in CESM as a consequence of thermal wind balance (Eq. 1).

**"Line 318; I do not follow why the smaller...":** *We guess that the reviewer may have misunderstood this paragraph since we did not intended to say that the difference of $T_y$ is a direct consequence of the slightly smaller size of the injected AM-SO$_4$ particles in CESM. However, we see that our formulation may have implied this. Additionally, we found that we messed something up in this sentence as the injected AM-SO$_4$ particles are larger in CESM than in ECHAM and not smaller as stated in the sentence. Therefore, we changed the text passage in the manuscript:*
In accordance with Niemeier et al. (2020), the sulfate burden for a given injection rate is substantially larger in CESM than in ECHAM, which is predominantly due to a stronger w* and the smaller size of the injected AM-SO₄ particles.

*Additionally, we included another Figure reference:*
Given the characteristic meridional distribution of sulfate particles for a 2point injection, this results in a higher sulfate burden in the subtropical stratosphere relative to the tropical one in CESM (Fig. 14 a).

**"Line 330; 'on' --> 'with'":** *We changed the sentence in the manuscript and additionally corrected an erroneous Figure reference in this sentence:*
Additionally, also the anomalies of $T_y$ and u agree reasonably well on with each other (Fig. 11 a,b) (Fig. 13 d,e).

**"Line 338; This is true only for...":** *We agree to the reviewer that this sentence only holds true for the lower-stratospheric QBO and changed the sentence in the manuscript:*
For an injection rate of 25Tg(S) yr$^{-1}$, the lower-stratospheric QBO is locked in a permanent westerly phase in ECHAM, while it is locked in a permanent easterly phase in CESM (Fig. 12 j, k) (Fig. 11 j, k).

**"Line 357; correspond to the positive anomalies...":** *The reviewer is right in that the lower-stratospheric negative ozone anomalies correlate with the lower-stratospheric positive sulfate mass mixing ratio, which is what we wanted to highlight by this sentence. However, given the fact that the*

*sulfate aerosols are the root cause for the ozone depletion, we assumed it to be unambiguous that we meant an anticorrelation at this point. Additionally, Figure 15 of the revised manuscript (Figure 14 of the preprint) doesn't show the anomaly of the sulfate mass mixing ratio as mentioned by the reviewer, but its absolute value, which can't be negative. However, we changed the manuscript to ease the understanding of this sentence:*

Thereby, the strength and the location of the negative ozone anomalies closely correspond to  the spatial distribution of the sulfate aerosols as represented by the sulfate mass mixing ratio.

**"Lines 363-368; Again, this is assuming a chain...":** *Following the reviewer, we have thought about this text passage very carefully and came to the same conclusion. We agree that we have been too speculative here and that we can't assume this strong causation chain at this point for CESM. The upper-stratospheric ozone heating rate anomalies above 20hPa are very small (not shown) and also the upper stratospheric temperature gradient anomalies, which we assumed to be caused by the upper-stratospheric ozone anomalies, are comparatively small. To be even more specific, those temperature gradient anomalies are roughly of the same magnitude as temperature anomalies induced by SMC anomalies, which can be seen in Figure 13 of the revised manuscript (Figure 12 of the preprint). Given the fact that thermal wind balance works either way, the upper-stratospheric westerly anomaly and its associated temperature gradient anomalies for CESM-region-so4-25 (Fig. 12 i of the revised manuscript, Fig. 11 i of the preprint) could also be caused by the lower-stratospheric easterly anomaly and its associated SMC. Therefore, we deleted the paragraph from Line 363-386 in the preprint and its not longer included in the revised manuscript:*

~~Additionally, the ozone concentration increases at altitudes around 5 to 10 hPa, which causes additional absorption of SW radiation and a positive heating rate around 5 hPa (Fig. 14). The accompanied slightly positive temperature anomaly causes the positive anomalies of $T_y$ close to the equator between 2.5 and 5 hPa (Fig. 12 i). Following thermal wind balance, this induces westerlies below, as observed between 5 and 10 hPa. The SMC and the stratospheric temperatures have to adopt to these westerlies, which causes the anomalies of $T_y$ to have the opposite sign than in ECHAM (Fig. 12 h). This ultimately explains the easterly anomalies below via changes in thermal wind balance.~~

*Additionally, we changed the paragraph, which followed Lines 363-368 of the preprint, accordingly (Lines 425-430 of the revised manuscript):*

We  argue that for the region injections  this partial compensation of aerosol-induced heating by ozone-induced cooling is likely to contribute significantly to the observed different anomalies of $T_y$ in the lower tropical stratosphere between 40hPa and 80hPa in CESM compared to ECHAM (Fig. 13 f–i).  Thereby, the aerosol-induced changes in the ozone concentration help to prevent the QBO from being locked in a permanent lower-stratospheric westerly phase like in ECHAM, which has no interactive ozone chemistry.

*We also changed the corresponding text passage in Section 5 accordingly:*

We think that this QBO response in CESM  partly is a result of local changes of the ozone concentration in the tropical stratosphere and its associated changes in SW heating.
The reduction of the SW heating in the lower stratosphere due to ozone depletion partly compensates the LW heating by sulfate particles in this region, which results in a meridionally more uniform temperature anomaly and, accordingly, in a weaker westerly anomaly above the injection layer following thermal wind balance.

**"Line 371; partial":** *Done.*

**"Lines 369-382; Another way to check...":** *We agree to the reviewer that this would be the ideal solution to investigate the impact of aerosol-induced ozone depletion on the QBO response in CESM and we also thought about performing this type of simulations. However, it is not as easy as it seems to turn off the interactive ozone chemistry in CESM2 as one may think and furthermore, we had limited computational capacities, which is why we have not been able to perform these simulations. Nevertheless, Richter et al. (2017) have exactly done this type of simulations for equatorial and extratropical single-point injections using CESM1, which we also underlined in Sec. 5 in Lines 520-525 of the revised manuscript (Lines 461-466 of the preprint). They concluded: "In short, interactive chemistry changes the heating and momentum budgets in the tropical stratosphere, causing notable changes to the QBO."*

**"Eqs R1-R2; For someone that isn't an atmospheric chemistry...":**
*We agree that the terms in equations R1 and R2 need some more explanation. Therefore, we have added some more details to the manuscript:*
The 2point simulations result in little changes in ozone concentration near the equator, due to lower $SO_4$ concentrations, while a decrease is observed in the subtropics and midlatitudes  (Fig. 15 a,c). This ozone decrease is mostly driven by the increase in the surface area density (SAD) produced by the aerosols, that enhances the ozone destruction by halogens (such as HOCl) in the heterogeneous reaction (at T <200K) $ClONO_2 + H_2O \rightarrow HOCl + HNO_3$ (R1) where $ClONO_2$ is one of the main stratospheric reservoir of inorganic chlorine (see Seinfeld and Pandis, 1998, Sec. 5.6).
(...)
The enhanced SAD results in a reduction in reacting nitrogen ($N_2O_5$, that is a catalyst for ozone destruction) due to the heterogeneous reaction (...)

**"Lines 419-426; Please see prior comments...":** *As we wrote earlier in our answer to reviewer 2 and in our answer to reviewer 1, we think that it is reasonable to assume causation in the lower stratosphere. There are no doubts that the aerosol-induced lower stratospheric temperature anomaly (see Fig. 2 of the revised manuscript) is the primary perturbation, to which the stratospheric wind field has to adopt in way thermal wind balance predicts. In order to highlight this further, we restructured Section 3 of the manuscript and included an extra Section on the stratospheric heating as the initial*

*perturbation (Section 3.1.1 of the revised manuscript), as mentioned in the beginning of our answers to the reviewer.*

*However, we followed the comment of reviewer 2 and formulated our argumentation more defensive at this point in order to take into account the concerns presented by all of the reviewers. We changed a passage of Section 5 in the revised manuscript as follows:*

– For the equatorial point and for the region injections the  aerosol warming peaks more or less sharply at the equator, causing a weakening of the poleward $T_y$ in the lower tropical stratosphere. This generates  a westerly wind anomaly and eventually forces the QBO into a permanent westerly phase.

– In contrast, $T_y$ is basically not modified in the tropics for the 2point injections due to a meridionally nearly uniform warming. Therefore, the QBO remains approximately in its natural state.

 Obviously, linking the QBO response to artificial sulfur injections to the meridional shape of the aerosol-induced temperature anomaly offers the possibility to explain the fundamentally different response of the QBO to all of our three injection strategies simulated with ECHAM in a stringent manner. This is a clear advancement compared to earlier studies, for example Niemeier and Schmidt (2017) or Tilmes et al. (2018), who did not adequately discussed differences in the QBO response between different injection strategies. The dependency of the QBO modification on the meridional structure of the lower stratospheric temperature anomaly via thermal wind balance may also be helpful to explain the significant acceleration of the QBO found by Richter et al. (2017) for extratropical single-point injections at 15°S, 15°N, 30°S, and 30°N, of which the causes were not finally determined.

Therewith, our results indicate that the modification of thermal wind balance in the lower tropical stratosphere between 40hPa and 80hPa is a simple but sufficient framework to explain the simulated differences in the QBO modification for our three tested injection strategies as it directly links the zonal wind anomaly to the $T_y$ anomaly. Furthermore, we have shown that the different QBO responses to the different injection strategies correspond to the observed differences in the acceleration of the residual tropical upwelling in ECHAM. For the point and the region injections, the residual tropical upwelling increases statistically significantly, while it does not increase for the 2point injections, which corresponds to the weak modification of the QBO observed for the 2point injections in ECHAM. ~~However, a comparison to ECHAM simulations with a permanent QBO easterly phase (Niemeier et al., 2011) indicate that a substantial part of the tropical upwelling anomaly can be attributed to the SMC of the modified QBO itself. The impact of the acceleration of the BDC as a whole, which is caused by an increased extratropical wave forcing as described in Tilmes et al. (2018) and Bittner et al. (2016), onto the QBO response may be rather small. Therefore, the increase of -ω*u and the associated slowdown of the QBO downward propagation maybe seen partly as a self-maintaining process of the perturbed QBO rather than its initial cause. Nevertheless, more research into the specific role of the SMC of the QBO~~

However, the reason for these differences is again the degree of tropical confinement of the aerosol-induced warming of the lower stratosphere because the modification of the residual circulation does ultimately also depend on the meridional gradient of the aerosol-induced temperature anomaly. This shows that using thermal wind balance as the overriding concept to explain the observed differences in the QBO response is reasonable.

An increase of the injection rate from 5 Tg(S) yr$^{-1}$ to 25 Tg(S) yr$^{-1}$ as well as an injection of AM−SO$_4$ instead of SO$_2$ act to strengthen the specific QBO response of all three injection strategies.  This has been shown to be a consequence of the stronger warming of the tropical lower stratosphere relative to the subtropical one for the point and region injections, which results in a stronger westerly anomaly.

**"Line 466; assess":** *Done.*

*Nevertheless, we still think that it is reasonable to argue the QBO changes via thermal wind balance since thermal wind balance and the residual circulation are inherently linked within the TEM framework. In fact, the residual circulation is a necessity to hold the zonal mean flow in thermal wind balance because otherwise the wave forcing would continuously drive it away from thermal wind balance (see e.g. Holton, 2004). Consequently, after a thermal perturbation such as aerosol heating, it's the residual circulation which changes and modifies the zonal mean flow in order to restore thermal wind balance. Therefore, we think that one can't take the modification of thermal wind balance and the increased residual tropical upwelling as two separate processes when investigating the QBO changes due to geoengineering. The increased tropical upwelling has to be rather interpreted as the mechanism*

*by which the system restores thermal wind balance within the tropics after the aerosol-induced temperature perturbation. This is highlighted in Section 3.1.3 of the revised manuscript:*

However, despite QBO changes due to artificial sulfur injections are frequently interpreted as the consequence of an increased residual tropical upwelling and a modification of thermal wind balance, one can't see both as two separate processes. In contrast, the acceleration of the BDC discussed in Section 3.1.2 is rather the mechanism by which thermal wind balance is reestablished for the aerosol-induced lower stratospheric temperature anomaly (Holton, 2004). While the differences in the increase of w* directly explain the different QBO response to different injection strategies in first order (see Sec. 3.1.2), in this Section we show that the differences in the QBO response can be also linked directly to the meridional shape of the aerosol-induced lower stratospheric temperature anomaly via thermal wind balance.

*One may now still argue that thermal wind balance is of inherent diagnostic nature and, therefore, does not allow for causal conclusions like we draw in Section 3.1.3 of the revised manuscript (Section 3.1 of the preprint); a concern which was put forth by reviewer 1 and 2. Nevertheless, for artificial sulfur injections, there are no doubts that the absorption of radiation in the near IR and terrestrial wavelengths by the sulfate aerosols and the associated positive temperature anomaly in the lower stratosphere are the initial and major perturbations within the lower tropical stratosphere (see Fig. 2 of the modified manuscript). In a second step this heating causes changes in dynamics. Since the final link between dynamics and aerosol-induced temperature anomalies is thermal wind balance, which is also acknowledged by reviewer 3 ("Although changes to thermal wind balance are there, (...)"), the lower stratospheric wind field has to adjust to this initial temperature perturbation in a way thermal wind balance states as it is an atmospheric state equation and, therefore, always holds on reasonable time scales, even in the tropics. Therefore, we think that it's reasonable to assume causation at this point and this was also done by Aquila et al. (2014).*

*Accepting that, thermal wind balance clearly states that the QBO response to artificial sulfur injections depends on the strength of the aerosol-induced modification of the meridional temperature gradient in the tropics, or – simply spoken – the strength of tropical confinement of the artificial aerosol cloud. This theory is clearly supported by our results, which also show a clear dependence of the QBO response to the strength of the associated modification of the meridional temperature gradient driven by the tropical confinement of the aerosol cloud (see Fig. 5 of the revised manuscript, Fig. 3 of the preprint). The absolute strength of the tropical temperature anomaly does not explain the observed differences in the QBO response between different injection strategies. We think this is a clear advancement compared to earlier geoengineering studies, which never clearly made this link to our knowledge. This is highlighted in Section 3.1.3 of the revised manuscript and was already included in Section 3.1 of the preprint:*

Our results clearly show that differences in the QBO response with respect to our three tested injection strategies are linked to differences in the meridional structure of the aerosol-induced temperature anomaly. Therefore, the absolute strength of the aerosol-induced lower stratospheric temperature anomaly does not permit a statement about the strength of the QBO modification when comparing different injection strategies. For instance, the tropical (i.e. 5°N to 5°S) mean temperature anomaly

within the injection layer is more than twice as high in 2point-so2-25 than in point-so2-5 (Fig. 6). However, the QBO is heavily perturbed in point-so2-5, while for 2point-so2-25 it remains nearly unchanged (Fig. 1 d,k).

*Finally, the strength of the residual upwelling anomaly is also directly linked to the tropical confinement of the artificial aerosol cloud as shown e.g. by Dunkerton (1983) in a study on effect of the Mt. Agung eruption on the QBO. Only a locally confined heating can alter the residual upwelling, which is also implied by the residual mean meridional streamfunction (Plumb & Bell, 1982). Hence, this explains why the tropical residual upwelling does not increase in the 2point injection scenario (Fig. 5c of the preprint). Therefore, we added the following in Section 3.1.2 of the revised manuscript:*

Within the TEM framework, the characteristics of the general acceleration of the BDC can be further directly linked to the tropical confinement of the aerosol-induced temperature anomaly, as shown by Dunkerton (1983) in a study on the effect of the 1963 eruption of Mt. Agung on the QBO. Following the equation of the residual mean meridional streamfunction (Plumb and Bell, 1982), only a tropically confined heating modifies the BDC, whereas a meridionally uniform heating has no effect. This explains why the residual tropical upwelling increases in point-so-25 and region-so2-25, but not in 2point-so2-25: While in point-so2-25 and region-so2-25 the aerosol heating has an equatorial peak and decreases rather sharply towards the extratropics, in 2point-so2-25 it is meridionally nearly uniform within the tropics (see Sec. 3.1.1).Consequently, it is ultimately the meridional shape of the aerosol-induced lower stratospheric temperature anomaly, or –simply spoken – the degree of tropical confinement of the artificial sulfate aerosols, what determines the QBO response to artificial sulfate injections.

*This dependence of the w\* anomaly on the meridional structure of the aerosol-induced temperature anomaly shows once more the link between w\* and thermal wind balance. However, since thermal wind balance links the zonal wind anomalies associated with the QBO response and the meridional shape of the aerosol-induced temperature anomaly directly, we decided to argue via thermal wind balance in the subsequent Sections of the revised manuscript, which we wrote in Section 3.1.3 of the revised manuscript:*

Since it is the degree of tropical confinement of the artificial sulfate aerosols what is ultimately decisive for the observed QBO response also when explaining the observed QBO changes solely as the consequence of an increased residual tropical upwelling, we will use thermal wind balance in our argumentation throughout this study as it directly links the observed QBO changes to the observed aerosol-induced temperature anomalies.

**Minor issues:**
**"1. Line 303: 'slight easterly anomaly up to -3m/s' – this could be natural variability...":** *In order to investigate the reviewer's comment, we have calculated the statistical significance of the zonal wind anomalies at the 95%-level using a student's t-test. It shows that the weak easterly anomaly centered at 5°S and 40hPa in Figure 13 c of the revised manuscript (Figure 12 c of the preprint) is only just*

*statistically significant and at the edge to natural variability based on our significance test. Therefore, we agree that the weak easterly anomaly centered at 7°S and 40hPa in Figure 13 c of the revised manuscript (Figure 12 c of the preprint) might also be caused by natural variability, although this would be at the very upper edge of natural variability. Consequently, we have modified the text passages in the new manuscript:*

However, for an injection rate of 25Tg(S) yr$^{-1}$ a slight easterly anomaly of up to -3m s$^{-1}$ has been noticed at approximately 40hPa and 5°S  (Fig. 13 c), which is at the edge of extreme natural variability based on a student's t-test.